# Statistical Analysis of Quantum State Learning Process in Quantum Neural Networks

**Hao-kai Zhang**[1,3]**, Chenghong Zhu**[2,3]**, Mingrui Jing**[2,3]**, Xin Wang**[2,3*]

[1] Institute for Advanced Study, Tsinghua University, Beijing 100084, China
[2] Thrust of Artificial Intelligence, Information Hub,
Hong Kong University of Science and Technology (Guangzhou), China
[3] Institute for Quantum Computing, Baidu Research, Beijing, China

## Abstract

Quantum neural networks (QNNs) have been a promising framework in pursuing near-term quantum advantage in various fields, where many applications can be viewed as learning a quantum state that encodes useful data. As a quantum analog of probability distribution learning, quantum state learning is theoretically and practically essential in quantum machine learning. In this paper, we develop a no-go theorem for learning an unknown quantum state with QNNs even starting from a high-fidelity initial state. We prove that when the loss value is lower than a critical threshold, the probability of avoiding local minima vanishes exponentially with the qubit count, while only grows polynomially with the circuit depth. The curvature of local minima is concentrated to the quantum Fisher information times a loss-dependent constant, which characterizes the sensibility of the output state with respect to parameters in QNNs. These results hold for any circuit structures, initialization strategies, and work for both fixed ansatzes and adaptive methods. Extensive numerical simulations are performed to validate our theoretical results. Our findings place generic limits on good initial guesses and adaptive methods for improving the learnability and scalability of QNNs, and deepen the understanding of prior information's role in QNNs.

## 1 Introduction

Recent experimental progress towards realizing quantum information processors [1–3] has fostered the thriving development of the emerging field of quantum machine learning (QML) [4–18], pursuing quantum advantages in artificial intelligence. Different QML algorithms have been proposed for various topics, e.g., quantum simulations [19–23], chemistry [24–28], quantum data compression [29, 30], generative learning [31, 32] and reinforcement learning [33], where quantum neural networks (QNNs) become a leading framework due to the hardware restriction from noisy intermediate scale quantum (NISQ) [34] devices. As quantum analogs of artificial neural networks, QNNs typically refer to parameterized quantum circuits which are trainable based on quantum measurement results.

However, QNNs face a severe scalability barrier which might prevent the realization of potential quantum advantages. A notorious example is the barren plateau phenomenon [35] which shows that the gradient of the loss function vanishes exponentially in the system size with a high probability for randomly initialized deep QNNs, giving rise to an exponential training cost. To address this issue, a variety of training strategies has been proposed, such as local loss functions [36], correlated parameters [37], structured architectures [38–40], good initial guesses [41, 42], initialization heuristics near the identity [43–46], adaptive methods [27] and layerwise training [47], etc. Nevertheless, there

---

*felixxinwang@hkust-gz.edu.cn

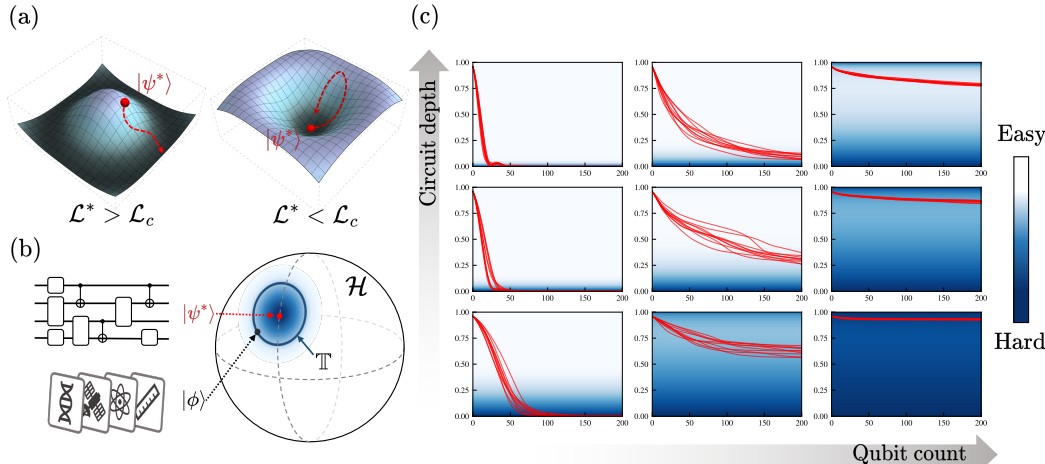

Figure 1: Sketches of our work characterizing the statistical performance of QNNs on quantum state learning tasks. (a) indicates the existence of a critical loss value $\mathcal{L}_c = 1 - 2^{-N}$ below which the local minima start to become severe to trap the training process. (b) depicts the setup of quantum state learning tasks where a QNN is used to learn an unknown target state encoding practical data. All target states with a constant distance to the output state $|\psi^*\rangle$ form our ensemble $\mathbb{T}$, depicted by the contour on the Bloch sphere. (c) shows typical loss curves for different qubit counts of $N = 2, 6, 10$ and circuit depths of $D = 1, 3, 5$. The intensity of the background colors represents the magnitude of our theoretical bound on the probability of encountering a local minimum, which hence signifies the hardness of optimization. One can find that the theoretical bound appropriately reflects the hardness encountered by the practical loss curves.

is a lack of scalability analyses for these strategies to guarantee their effectiveness. Especially, since the identity initialization and adaptive or layerwise training methods do not require uniformly random initialization of deep QNNs, they are out of the scope of barren plateaus and urgently need a comprehensive theoretical analysis to ascertain their performance under general conditions.

In this work, we analyze the learnability of QNNs from a statistical perspective considering the information of loss values. Specifically, given a certain loss value during the training process, we investigate the statistical properties of surrounding training landscapes. Here we mainly focus on quantum state learning tasks [48, 49], which can be seen as a quantum analog of probability distribution learning and play a central role in QML. To summarize, our contributions include:

- We prove a no-go theorem stating that during the process of learning an unknown quantum state with QNNs, the probability of avoiding local minima is of order $\mathcal{O}(N^2 2^{-N} D^2/\epsilon^2)$ as long as the loss value is lower than a critical threshold (cf. Fig. 1). The bound vanishes exponentially in the qubit count $N$ while only increases polynomially with the circuit depth $D$. The curvature of local minima is concentrated to the quantum Fisher information times a loss-dependent constant. The proof is mainly based on the technique of "subspace Haar integration" we developed in Appendix A.1. A generalized version for the local loss function is provided in Appendix C.

- We conduct extensive numerical experiments to verify our theoretical findings. We first compare our bound with practical loss curves to show the prediction ability on the statistical behavior of the actual training process. Then we sample landscape profiles to visualize the existence of asymptotic local minima. Finally, we compute the gradients and diagonalize the Hessian matrices to directly verify the correctness of our bound.

- Our results place general limits on the learnability of QNNs, especially for the training strategies beyond the scope of barren plateaus, including high-fidelity initial guesses, initialization heuristics near the identity, and adaptive and layerwise training methods. Moreover, our results provide a theoretical basis for the necessity of introducing prior information into QNN designs and hence draw a guideline for future QNN developments.

## 1.1 Related works

The barren plateau phenomenon was first discovered by [35], which proves that the variance of the gradient vanishes exponentially with the system size if the randomly initialized QNN forms a unitary 2-design. Thereafter, [36] finds the dependence of barren plateaus on the circuit depth for loss functions with local observables. [50] proves that training QNNs is in general NP-hard. [51] introduces barren plateaus from uncertainty which precludes learning scramblers. [51, 52] establish connections among the expressibility, generalizability and trainability of QNNs. [53] and [54] show that apart from barren plateaus, QNNs also suffer from local minima in certain cases.

On the other hand, many training strategies have been proposed to address barren plateaus. Here we only list a small part relevant to our work. [43–46] suggest that initializing the QNN near the identity could reduce the randomness and hence escape from barren plateaus. [27] and [47] propose adaptive and layerwise training methods which avoid using randomly initialized QNNs in order to avoid barren plateaus, whereas [55] finds counterexamples where the circuit training terminates close to the identity and remains near to the identity for subsequently added layers without effective progress.

## 2 Quantum computing basics and notations

We use $\| \cdot \|_p$ to denote the $l_p$-norm for vectors and the Schatten-$p$ norm for matrices. $A^\dagger$ is the conjugate transpose of matrix $A$. $\mathrm{tr}\, A$ represent the trace of $A$. The $\mu$-th component of the vector $\boldsymbol{\theta}$ is denoted as $\theta_\mu$ and the derivative with respect to $\theta_\mu$ is simply denoted as $\partial_\mu = \frac{\partial}{\partial \theta_\mu}$. We employ $\mathcal{O}$ as the asymptotic notation of upper bounds.

In quantum computing, the basic unit of quantum information is a quantum bit or qubit. A single-qubit pure state is described by a unit vector in the Hilbert space $\mathbb{C}^2$, which is commonly written in Dirac notation $|\psi\rangle = \alpha|0\rangle + \beta|1\rangle$, with $|0\rangle = (1,0)^T$, $|1\rangle = (0,1)^T$, $\alpha, \beta \in \mathbb{C}$ subject to $|\alpha|^2 + |\beta|^2 = 1$. The complex conjugate of $|\psi\rangle$ is denoted as $\langle\psi| = |\psi\rangle^\dagger$. The Hilbert space of $N$ qubits is formed by the tensor product "$\otimes$" of $N$ single-qubit spaces with dimension $d = 2^N$. We denote the inner product of two states $|\phi\rangle$ and $|\psi\rangle$ as $\langle\phi|\psi\rangle$ and the overlap is defined as $|\langle\phi|\psi\rangle|$. General mixed quantum states are represented by the density matrix, which is a positive semidefinite matrix $\rho \in \mathbb{C}^{d \times d}$ subject to $\mathrm{tr}\, \rho = 1$. Quantum gates are unitary matrices, which transform quantum states via the matrix-vector multiplication. Common single-qubit rotation gates include $R_x(\theta) = e^{-i\theta X/2}$, $R_y(\theta) = e^{-i\theta Y/2}$, $R_z(\theta) = e^{-i\theta Z/2}$, which are in the matrix exponential form of Pauli matrices

$$X = \begin{pmatrix} 0 & 1 \\ 1 & 0 \end{pmatrix}, \qquad Y = \begin{pmatrix} 0 & -i \\ i & 0 \end{pmatrix}, \qquad Z = \begin{pmatrix} 1 & 0 \\ 0 & -1 \end{pmatrix}. \tag{1}$$

Common two-qubit gates include controlled-X gate $\mathrm{CNOT} = I \oplus X$ ($\oplus$ is the direct sum) and controlled-Z gate $\mathrm{CZ} = I \oplus Z$, which can generate quantum entanglement among qubits.

### 2.1 Framework of Quantum Neural Networks

Quantum neural networks (QNNs) typically refer to parameterized quantum circuits $\mathbf{U}(\boldsymbol{\theta})$ where the parameters $\boldsymbol{\theta}$ are trainable based on the feedback from quantum measurement results using a classical optimizer. By assigning some loss function $\mathcal{L}(\boldsymbol{\theta})$, QNNs can be used to accomplish various tasks just like artificial neural networks. A general form of QNNs reads $\mathbf{U}(\boldsymbol{\theta}) = \prod_{\mu=1}^{M} U_\mu(\theta_\mu) W_\mu$, where $U_\mu(\theta_\mu) = e^{-i\Omega_\mu \theta_\mu}$ is a parameterized gate such as single-qubit rotations with $\Omega_\mu$ being a Hermitian generator. $W_\mu$ is a non-parameterized gate such as CNOT and CZ. The product $\prod_{\mu=1}^{M}$ is by default in the increasing order from the right to the left. $M$ denotes the number of trainable parameters. Note that QNNs with intermediate classical controls [56] can also be included in this general form theoretically. Commonly used templates of QNNs include the hardware efficient ansatz [26], the alternating-layered ansatz (ALT) [57] and the tensor-network-based ansatz [36, 58], which are usually composed of repeated layers. The number of repeated layers is called the depth of the QNN, denoted as $D$. Fig. 2 depicts an example of the ALT circuit. The gradients of loss functions of certain QNNs are evaluated by the parameter-shift rule [59–61] on real quantum devices. Hence we can train QNNs efficiently with gradient-based optimizers [62].

Here we focus on quantum state learning tasks, the objective of which is to learn a given target state $|\phi\rangle$ encoding practical data via minimizing the distance between the target state $|\phi\rangle$ and the output

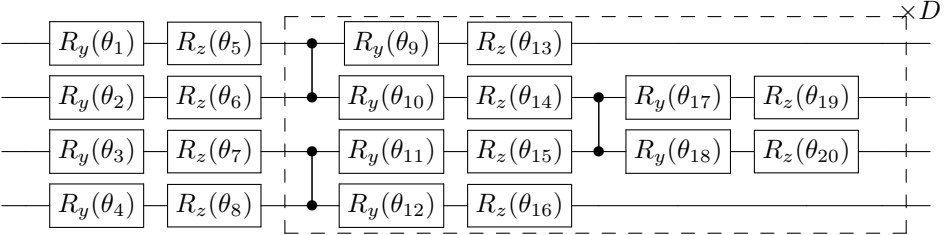

Figure 2: The quantum circuit of the alternating-layered ansatz on 4 qubits. The circuit starts with a $R_y$ layer and a $R_x$ layer, followed by $D$ repeated layers, where each layer contains alternating 2-qubit unit blocks of a CZ gate, two $R_y$ gates and two $R_z$ gates.

state from the QNN $|\psi(\boldsymbol{\theta})\rangle = U(\boldsymbol{\theta})|0\rangle^{\otimes N}$. The corresponding loss function is usually chosen as the fidelity distance

$$\mathcal{L}(\boldsymbol{\theta}) = 1 - |\langle\phi|\psi(\boldsymbol{\theta})\rangle|^2. \tag{2}$$

which can be efficiently calculated on quantum computers using the swap test [63]. An important quantity we used below characterizing the sensitivity of the QNN output state $|\psi(\boldsymbol{\theta})\rangle$ regarding its parameters $\boldsymbol{\theta}$, the quantum Fisher information (QFI) matrix $\mathcal{F}_{\mu\nu}$ [64], which is defined as the Riemannian metric induced from the fidelity distance (cf. Appendix A.4)

$$\mathcal{F}_{\mu\nu}(\boldsymbol{\theta}) = 2\,\mathrm{Re}\left[\langle\partial_\mu\psi|\partial_\nu\psi\rangle - \langle\partial_\mu\psi|\psi\rangle\langle\psi|\partial_\nu\psi\rangle\right]. \tag{3}$$

If the QFI is not full-rank, we say $|\psi(\boldsymbol{\theta})\rangle$ is over-parameterized [65, 66] meaning that there are redundant degrees of freedom in the parameters over the manifold dimension of the state.

## 3 Statistical characterization of quantum state learning in QNNs

In this section, we develop a no-go theorem characterizing the limitation of quantum neural networks in state learning tasks from a statistical perspective. In short, we prove that the probability of avoiding local minima during the process of learning an unknown quantum state with a QNN is of order $\mathcal{O}(2^{-N}M^2/\epsilon^2)$, where $N$ is the number of qubits, $M$ is the number of trainable parameters and $\epsilon$ represents the typical precision of measurements. The detailed upper bound also depends on the overlap between the value of the loss function and the QFI of the QNN. Our bounds significantly improve existing results of the trainability analysis of QNNs, which mainly focus on the randomness from the initialization and neglect the information of the loss function value. We will first introduce our ensemble setting in Section 3.1, then present our main theorem on local minima in Section 3.2 and finally show some results beyond local minima in Section 3.4.

### 3.1 Ensemble of the unknown target state

We first introduce the probability measure used in this work. The randomness studied by most of the previous work on the trainability analyzes of QNNs originates from the random initialization of trainable parameters [35], which usually depends on the circuit depth, specific choices of the QNN architecture and initialization strategies [44]. Meanwhile, the randomness can also come from the lack of prior information, such as learning an unknown quantum state or an unknown scrambler like a black hole [67]. We focus on the latter in the present work.

The usage of adaptive methods is usually not covered by common trainability analyses, however, the training process often tends to stagnate. With the aim of investigating the trainability at a specific loss function value, the ensemble is constructed by a uniform measure of overall pure states that have the same overlap with the current output state of the QNN. Specifically, suppose $\boldsymbol{\theta}^*$ is the current value of the trainable parameters. The overlap, or fidelity, between the output state $|\psi^*\rangle = |\psi(\boldsymbol{\theta}^*)\rangle$ and the target state $|\phi\rangle$ equals to $|\langle\phi|\psi^*\rangle| = p$. Thus, the target state can be decomposed as

$$|\phi\rangle = p|\psi^*\rangle + \sqrt{1-p^2}|\psi^\perp\rangle, \tag{4}$$

where $|\psi^\perp\rangle$ represents the unknown component in the target state $|\phi\rangle$ orthogonal to the learnt component $|\psi^*\rangle$. If no more prior information is known about the target state $|\phi\rangle$ except for the overlap

$p$, in the spirit of Bayesian statistics, $|\psi^\perp\rangle$ is supposed to be a random state uniformly distributed in the orthogonal complement of $|\psi^*\rangle$, denoted as $\mathcal{H}^\perp$. Such a Haar-random state can induce an ensemble of the unknown target state via Eq. (4), which we denote as $\mathbb{T} = \{|\phi\rangle \mid |\psi^\perp\rangle \text{ is Haar-random in } \mathcal{H}^\perp\}$. Graphically, $\mathbb{T}$ can be understood as a contour on the Bloch sphere of an $N$-qubit system with a constant distance to $|\psi^*\rangle$ as shown in Fig. 1(b). We remark that $\boldsymbol{\theta}^*$ can be interpreted as either an initial guess or an intermediate value during the training process so that our following results can be applied to the entire process of learning a quantum state. See Appendix A.1 for more details on the ensemble setting.

## 3.2 Exponentially likely local minima

We now investigate the statistical properties of the gradient $\nabla\mathcal{L}$ and the Hessian matrix $H_\mathcal{L}$ of the loss function $\mathcal{L}(\boldsymbol{\theta}^*)$ at the parameter point $\boldsymbol{\theta} = \boldsymbol{\theta}^*$ regarding the ensemble $\mathbb{T}$, and hence derive an upper bound of the probability that $\boldsymbol{\theta}^*$ is not a local minimum. For simplicity of notation, we represent the value of a certain function at $\boldsymbol{\theta} = \boldsymbol{\theta}^*$ by appending the superscript "$*$", e.g., $\nabla\mathcal{L}|_{\boldsymbol{\theta}=\boldsymbol{\theta}^*}$ as $\nabla\mathcal{L}^*$ and $H_\mathcal{L}|_{\boldsymbol{\theta}=\boldsymbol{\theta}^*}$ as $H_\mathcal{L}^*$. We define that $\boldsymbol{\theta}^*$ is a local minimum up to a fixed precision $\epsilon = (\epsilon_1, \epsilon_2)$ if and only if each of the gradient components is not larger than $\epsilon_1$ and the minimal eigenvalue of the Hessian matrix is not smaller than $-\epsilon_2$, i.e.,

$$\text{LocalMin}(\boldsymbol{\theta}^*, \epsilon) = \bigcap_{\mu=1}^{M} \{|\partial_\mu\mathcal{L}^*| \le \epsilon_1\} \ \cap \ \{H_\mathcal{L}^* \succ -\epsilon_2 I\}. \tag{5}$$

If $\epsilon_1$ and $\epsilon_2$ both take zero, Eq. (5) is reduced back to the common exact definition of the local minimum. However, noises and uncertainties from measurements on real quantum devices give rise to a non-zero $\epsilon$, where the estimation cost scales as $\mathcal{O}(1/\epsilon^\alpha)$ for some power $\alpha$ [68]. Specially, if $|\psi(\boldsymbol{\theta}^*)\rangle$ approaches the true target state $|\phi\rangle$ such that $\mathcal{L}^* \to 0$, we say $\boldsymbol{\theta}^*$ is a global minimum. That is to say, here "local minima" are claimed with respect to the entire Hilbert space instead of training landscapes created by different ansatzes. The expectation and variance of the first and second-order derivatives of the loss function are calculated and summarized in Lemma 1, with the detailed proof in Appendix B utilizing the technique we dubbed "subspace Haar integration" in Appendix A.1.

**Lemma 1** *The expectation and variance of the gradient $\nabla\mathcal{L}$ and Hessian matrix $H_\mathcal{L}$ of the fidelity loss function $\mathcal{L}(\boldsymbol{\theta}) = 1 - |\langle\phi|\psi(\boldsymbol{\theta})\rangle|^2$ at $\boldsymbol{\theta} = \boldsymbol{\theta}^*$ with respect to the target state ensemble $\mathbb{T}$ satisfy*

$$\mathbb{E}_\mathbb{T}[\nabla\mathcal{L}^*] = 0, \quad \text{Var}_\mathbb{T}[\partial_\mu\mathcal{L}^*] = f_1(p,d)\mathcal{F}_{\mu\mu}^*. \tag{6}$$

$$\mathbb{E}_\mathbb{T}[H_\mathcal{L}^*] = \frac{dp^2-1}{d-1}\mathcal{F}^*, \quad \text{Var}_\mathbb{T}[\partial_\mu\partial_\nu\mathcal{L}^*] \le f_2(p,d)\|\Omega_\mu\|_\infty^2\|\Omega_\nu\|_\infty^2. \tag{7}$$

*where $\mathcal{F}$ denote the QFI matrix in Eq. (3) and $\Omega_\mu$ is the generator of the gate $U_\mu(\theta_\mu)$. $f_1$ and $f_2$ are functions of the overlap $p$ and the Hilbert space dimension $d$, i.e.,*

$$f_1(p,d) = \frac{p^2(1-p^2)}{d-1}, \quad f_2(p,d) = \frac{32(1-p^2)}{d-1}\left[p^2 + \frac{2(1-p^2)}{d}\right]. \tag{8}$$

The exponentially vanishing variances in Lemma 1 imply that the gradient and Hessian matrix concentrate to their expectations exponentially in the number of qubits $N$ due to $d = 2^N$ for a $N$-qubit system. Thus the gradient concentrates to zero and the Hessian matrix concentrates to the QFI $\mathcal{F}^*$ times a non-vanishing coefficient proportional to $(p^2 - 1/d)$. Since the QFI is always positive semidefinite, the expectation of the Hessian matrix is either positive semidefinite $\mathcal{L}^* = 1 - p^2 < 1 - 1/d$, or negative semidefinite if $\mathcal{L}^* > 1 - 1/d$, as illustrated in Fig. 1(a). The critical point $\mathcal{L}_c = 1 - 1/d$ coincides with the average fidelity distance of two Haar-random pure states, which means that as long as $|\psi^*\rangle$ has a higher fidelity than the average level of all states, the expectation of the Hessian matrix would be positive semidefinite.

Using Lemma 1, we establish an exponentially small upper bound on the probability that $\boldsymbol{\theta}^*$ is not a local minimum in the following Theorem 2, where the generator norm vector $\boldsymbol{\omega}$ is defined as $\boldsymbol{\omega} = (\|\Omega_1\|_\infty, \|\Omega_2\|_\infty, \ldots, \|\Omega_M\|_\infty)$.

**Theorem 2** *If the fidelity loss function satisfies $\mathcal{L}(\boldsymbol{\theta}^*) < 1 - 1/d$, the probability that $\boldsymbol{\theta}^*$ is not a local minimum of $\mathcal{L}$ up to a fixed precision $\epsilon = (\epsilon_1, \epsilon_2)$ with respect to the target state ensemble $\mathbb{T}$ is upper bounded by*

$$\Pr_{\mathbb{T}} \left[ \neg \operatorname{LocalMin}(\boldsymbol{\theta}^*, \epsilon) \right] \leq \frac{2f_1(p, d) \|\boldsymbol{\omega}\|_2^2}{\epsilon_1^2} + \frac{f_2(p, d) \|\boldsymbol{\omega}\|_2^4}{\left( \frac{dp^2 - 1}{d - 1} e^* + \epsilon_2 \right)^2}, \tag{9}$$

*where $e^*$ denotes the minimal eigenvalue of the QFI matrix at $\boldsymbol{\theta} = \boldsymbol{\theta}^*$. $f_1$ and $f_2$ are defined in Eq. (8) which vanish at least of order $1/d$.*

A sketch version of the proof is as follows, with the details in Appendix B. By definition in Eq. (5), the left-hand side of Eq. (9) can be upper bounded by the sum of two terms: the probability that one gradient component is larger than $\epsilon_1$, and the probability that the Hessian matrix is not positive definite up to $\epsilon_2$. The first term can be bounded by Lemma 1 and Chebyshev's inequality, i.e.,

$$\Pr_{\mathbb{T}} \left[ \bigcup_{\mu=1}^{M} \{ |\partial_\mu \mathcal{L}^*| > \epsilon_1 \} \right] \leq \sum_{\mu=1}^{M} \frac{\operatorname{Var}_{\mathbb{T}}[\partial_\mu \mathcal{L}^*]}{\epsilon_1^2} = \frac{f_1(p, d)}{\epsilon_1^2} \operatorname{tr} \mathcal{F}^*, \tag{10}$$

where the QFI diagonal element is bounded as $\mathcal{F}_{\mu\mu} \leq 2\|\Omega_\mu\|_\infty^2$ by definition and thus $\operatorname{tr} \mathcal{F}^* \leq 2\|\boldsymbol{\omega}\|_2^2$. After assuming $p^2 > 1/d$, the second term can be upper bounded by perturbing $\mathbb{E}_{\mathbb{T}}[H_{\mathcal{L}}^*]$ to obtain a sufficient condition of positive definiteness (see Appendix A.2), and then utilizing Lemma 1 with the generalized Chebyshev's inequality for matrices (see Appendix A.3), i.e.,

$$\Pr_{\mathbb{T}} \left[ H_{\mathcal{L}}^* \not\succ -\epsilon_2 I \right] \leq \sum_{\mu,\nu=1}^{M} \frac{\operatorname{Var}_{\mathbb{T}} \left[ \partial_\mu \partial_\nu \mathcal{L}^* \right]}{\left( \frac{dp^2 - 1}{d - 1} e^* + \epsilon_2 \right)^2} \leq \frac{f_2(p, d) \|\boldsymbol{\omega}\|_2^4}{\left( \frac{dp^2 - 1}{d - 1} e^* + \epsilon_2 \right)^2}. \tag{11}$$

Combining the bounds regarding the gradient and hessian matrix, one arrives at Eq. (9). ∎

Theorem 2 directly points out that if the loss function takes a value lower than the critical threshold $\mathcal{L}_c = 1 - 1/d$, then the surrounding landscape would be a local minimum for almost all of the target states, the proportion of which is exponentially close to 1 as the qubit count $N = \log_2 d$ grows. Note that $\|\boldsymbol{\omega}\|_2^2 = \sum_{\mu=1}^{M} \|\Omega_\mu\|_\infty^2$ scales linearly with the number of parameters $M$ and at most polynomially with the qubit count $N$. Because practically the operator norm $\|\Omega_\mu\|_\infty$ is constant such as the generators of Pauli rotations $\|X\|_\infty = \|Y\|_\infty = \|Z\|_\infty = 1$, or grows polynomially with the system size such as the layer with globally correlated parameters [56] and the global evolution in analog quantum computing [69]. Here we focus on the former and conclude that the upper bound in Theorem 2 is of order $\mathcal{O}(2^{-N} M^2 / \epsilon^2)$, implying the exponential training cost. The conclusion also holds for noisy quantum states (see Appendix B). A similar result for the so-called local loss function, like the energy expectation used in variational quantum eigensolvers, is provided in Appendix C.

In principle, if one could explore the whole Hilbert space with exponentially many parameters, $\boldsymbol{\theta}^*$ was a saddle point at most instead of a local minimum since there must exist a unitary connecting the learnt state $|\psi^*\rangle$ and the target state $|\phi\rangle$. This is also consistent with our bound by taking $M \in \Omega(2^{N/2})$ to cancel the $2^{-N}$ factor such that the bound is no more exponentially small. However, the number of parameters one can control always scales polynomially with the qubit count due to the memory constraint. This fact indicates that if the QNN is not designed specially for the target state using some prior knowledge so that the "correct" direction towards the target state is contained in the accessible tangent space, the QNN will have the same complexity as the normal quantum state tomography.

**Dependence on the loss value $\mathcal{L}^*$.** The dependence of the bound in Theorem 2 on the overlap $p = \sqrt{1 - \mathcal{L}^*}$ shows that, as the loss function value becomes lower, the local minima becomes denser so that the training proceeds harder. This agrees with the experience that the loss curves usually decay fast at the beginning of a training process and slow down till the convergence. If $e^* \neq 0$, the second term in Eq. (9) becomes larger as $p^2 \to 1/d$, suggesting that the local minima away from the critical point $\mathcal{L}_c$ is more severe than that near $\mathcal{L}_c$. Moreover, if $\epsilon_2 = 0$, the bound diverges as the QFI minimal eigenvalue $e^*$ vanishes, which reflects the fact that over-parameterized QNNs have many equivalent local minima connected by the redundant degrees of freedom of parameters.

By contrast, if $\mathcal{L}^* > \mathcal{L}_c$, the results could be established similarly by slightly modifying the proof yet with respect to local maxima, as depicted in Fig. 1(a). However, the critical point $\mathcal{L}_c = 1 - 2^{-N}$

moves to 1 exponentially fast as $N$ increases, i.e., the range of $\mathcal{L}^*$ without severe local minima shrink exponentially. Hence for large-scale systems, even with a polynomially small fidelity, one would encounter a local minimum almost definitely if no more prior knowledge can be used.

### 3.3 Implication on the learnability of QNNs

In practical cases, if the QNN is composed of $D$ repeated layers with $z$ trainable parameters for each layer and each qubit, then the total number of trainable parameters becomes $M = NDz$, and hence the probability of avoiding local minima is of order $\mathcal{O}(N^2 2^{-N} D^2/\epsilon^2)$, which increases quadratically as the QNN becomes deeper. This seems contrary to the conclusion from barren plateaus [35] where deep QNNs lead to poor trainability. But in fact, they are complementary to each other. The reason is that the ensemble here originates from the unknown target state instead of the random initialization. Similar to classical neural networks, a deeper QNN has stronger expressibility, which creates a larger accessible manifold to approach the unknown state and may turn a local minimum into a saddle point with the increased dimensions. But on the other hand, a deeper QNN with randomly initialized parameters leads to barren plateaus [36]. In short, the local minima here arise due to the limited expressibility together with a non-vanishing fidelity while barren plateaus stem from the strong expressibility together with the random initialization.

To solve this dilemma, our results suggest that a well-designed QNN structure taking advantage of prior knowledge of the target state is vitally necessary. Otherwise, a good initial guess (i.e., an initial state with high fidelity) solely is hard to play its role. An example of prior knowledge from quantum many-body physics is the tensor network states [70] satisfying the entanglement area law, which lives only in a polynomially large space but generally can not be solved in two and higher spatial dimensions by classical computers. Other examples include the UCCSD ansatz [71] in quantum chemistry and the QAOA ansatz [72] in combinatorial optimization, which all attempt to utilize the prior knowledge of the target states.

Finally, we remark that our results also place general theoretical limits for adaptive [27] or layer-wise training methods [47]. Relevant phenomena are observed previously in special examples [55]. Adaptive methods append new training layers incrementally during the optimization instead of placing a randomly initialized determinate ansatz at the beginning, which is hence beyond the scope of barren plateaus [35]. Nevertheless, our results imply that for moderately large systems with $\mathcal{L}^* < \mathcal{L}_c$, every time a new training layer is appended, the learnt state $|\psi^*\rangle$ would be a local minimum of the newly created landscape so that the training process starting near $|\psi^*\rangle$ would go back to the original state $|\psi^*\rangle$ without any effective progress more than applying an identity. Note that in adaptive methods, one usually initializes the new appended layer near the identity to preserve the historical learnt outcomes. Similar phenomena are also expected to occur in the initialization strategies where the circuit begins near the identity [43, 44, 46]. We emphasize that our results do not imply the ineffectiveness of all adaptive methods. Instead, they only suggest that simplistic brute-force adaptive methods provide no significant benefit in terms of enhancing learnability on average.

### 3.4 Concentration of training landscapes

Theorem 2 analyses the statistical properties of the vicinity of a certain point $\boldsymbol{\theta}^*$, i.e., the probability distributions of the gradient and Hessian matrix of $\boldsymbol{\theta}^*$. To characterize the training landscape beyond the vicinity, a pointwise result is established in Proposition 3 with the proof in Appendix B.

**Proposition 3** *The expectation and variance of the fidelity loss function $\mathcal{L}$ with respect to the target state ensemble $\mathbb{T}$ can be exactly calculated as*

$$
\begin{aligned}
\mathbb{E}_{\mathbb{T}}\left[\mathcal{L}(\boldsymbol{\theta})\right] &= 1 - p^2 + \frac{dp^2 - 1}{d - 1}g(\boldsymbol{\theta}), \\
\mathrm{Var}_{\mathbb{T}}\left[\mathcal{L}(\boldsymbol{\theta})\right] &= \frac{1 - p^2}{d - 1}g(\boldsymbol{\theta})\left[4p^2 - \left(2p^2 - \frac{(d-2)(1-p^2)}{d(d-1)}\right)g(\boldsymbol{\theta})\right],
\end{aligned}
\tag{12}
$$

*where $g(\boldsymbol{\theta}) = 1 - |\langle\psi^*|\psi(\boldsymbol{\theta})\rangle|^2$.*

Since the factor $g(\boldsymbol{\theta})$ takes its global minimum at $\boldsymbol{\theta}^*$ by definition, the exponentially small variance in Proposition 3 implies that the entire landscape concentrates exponentially in the qubit count to

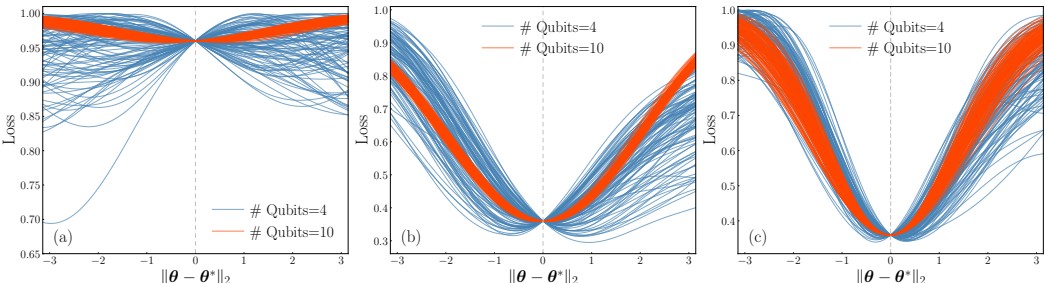

Figure 3: Samples of training landscapes along randomly chosen directions as a function of the distance $\|\boldsymbol{\theta} - \boldsymbol{\theta}^*\|$ for random target states from $\mathbb{T}$ with the sample size 200, the qubit count $N = 4$ and $N = 10$ and the overlap (a) $p = 0.2$ and (b) $p = 0.8$, respectively. The setup in (c) is the same as in (b) but fixes the target state and only samples the directions. A clear convex shape is present in the samples from $N = 10$ but absent in the samples from $N = 4$. This phenomenon intuitively shows that the training landscapes from the ensemble $\mathbb{T}$ are concentrated to a local minimum around $\boldsymbol{\theta}^*$ as the qubit count increases.

the expectation $\mathbb{E}_{\mathbb{T}}\left[\mathcal{L}(\boldsymbol{\theta})\right]$ with a pointwise convergence (not necessarily a uniform convergence), which takes its global minimum at $\boldsymbol{\theta}^*$ with respect to the training landscape as long as $p^2 > 1/d$. For QNNs satisfying the parameter-shift rule, the factor $g(\boldsymbol{\theta})$ along the Cartesian axis corresponding to $\theta_\mu$ passing through $\boldsymbol{\theta}^*$ will take the form of a trigonometric function $\frac{1}{2}\mathcal{F}_{\mu\mu}^* \sin^2(\theta_\mu - \theta_\mu^*)$, which is elaborated in Appendix B. Other points apart from $\boldsymbol{\theta}^*$ is allowed to have a non-vanishing gradient expectation in our setup, which leads to prominent local minima instead of plateaus [35].

## 4 Numerical experiments

Previous sections theoretically characterize the limitation of QNNs in state learning tasks considering the information of the loss value $\mathcal{L}^*$. In this section, we verify these results by conducting numerical experiments on the platform Paddle Quantum [73] and Tensorcircuit [74] from the following three perspectives. The codes for numerical experiments can be found in [75].

**Comparison with loss curves.** Firstly, we show the prediction ability of Theorem 2 by direct comparison with experimental loss curves in Fig. 1(c). We create 9 ALT circuits for qubit counts of $2, 6, 10$ and circuit depth of $1, 3, 5$ with randomly initialized parameters, denoted as $\boldsymbol{\theta}^*$. For each circuit, we sample 10 target states from the ensemble $\mathbb{T}$ with $p = 0.2$ and then generate 10 corresponding loss curves using the Adam optimizer with a learning rate 0.01. We exploit the background color intensity to represent the corresponding bounds from Theorem 2 by assigning $e^* = 0.1$ and $\epsilon_1 = \epsilon_2 = 0.05$. One can find that the loss curves decay fast at the beginning and then slow down till convergence, in accordance with the conclusion that the probability of encountering local minima is larger near the bottom. The convergent loss value becomes higher as the qubit count grows and can be partially reduced by increasing the circuit depth, which is also consistent with Theorem 2.

**Landscape profile sampling.** We visualize the existence of asymptotic local minima by sampling training landscape profiles in Fig. 3. Similar to the setup above, we create an ALT circuit with randomly initialized parameters $\boldsymbol{\theta}^*$, sample 200 target states from the ensemble $\mathbb{T}$ and compute the loss values near $\boldsymbol{\theta}^*$. Figs. 3(a) and (b) are obtained by randomly choosing a direction for each landscape sample, while Fig. 3(c) is obtained by randomly sampling 200 directions for one fixed landscape sample. There is no indication of local minimum in the case of few qubits and small fidelity as shown by the blue curves in Fig. 3(a). However, as the qubit number grows, the landscape profiles concentrate into a clear convex shape centered at $\boldsymbol{\theta}^*$ for both $p = 0.2$ and $p = 0.8$, where the curvature for $p = 0.8$ is larger due to the factor $(p^2 - 1/d)$ in Eq. (7). Fig. 3(c) further demonstrates that beyond the convexity along a specific direction, $\boldsymbol{\theta}^*$ is a highly probable local minimum in the case of large qubit counts.

**Probability evaluation.** Finally, we compute the gradients and diagonalize the Hessian matrices to directly verify the exponentially likely local minima proposed by Theorem 2 in Fig. 4. Similar to

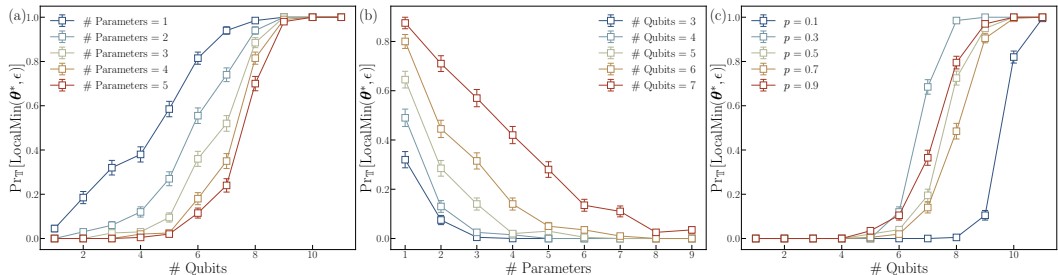

Figure 4: Numerical evaluation for the probability that $\boldsymbol{\theta}$ is a local minimum up to a fixed precision $\epsilon$, i.e., $\mathrm{Pr}_{\mathbb{T}}\left[\mathrm{LocalMin}(\boldsymbol{\theta}^*, \epsilon)\right]$ for different qubit count, the number of trainable parameters and the overlap $p$, with the error bar representing the statistical uncertainty in experiments. (a) shows that the probability converges to 1 rapidly with the increasing qubit count. (b) shows that the probability is reduced by increasing the number of parameters, implying the local minimum phenomenon is mitigated. (c) illustrates that the probability of encountering local minima always converges to 1 for any fixed overlap $p$. $p = 0.8$ for both (a) and (b) and the number of parameters in (c) is 6.

the setup above, we create ALT circuits for qubit count from $N = 1$ to 11 with depth $D = 5$ and sample 200 target states from $\mathbb{T}$ for each circuit. After specifying a certain subset of parameters to be differentiated, we estimate the probability that $\boldsymbol{\theta}$ is a local minimum by the proportion of samples satisfying the condition $\mathrm{LocalMin}(\boldsymbol{\theta}^*, \epsilon)$, where we assign $\epsilon_1 = \epsilon_2 = 0.05$. One can find that the probability of encountering local minima saturates to 1 very fast as the qubit count increases for arbitrary given values of $p$, and at the same time, it can be reduced by increasing the number of trainable parameters, which is consistent with the theoretical findings in Theorem 2.

## 5    Conclusion and outlook

In this paper, we prove that during the process of learning an unknown quantum state with QNNs, the probability of avoiding local minima is of order $\mathcal{O}(N^2 2^{-N} D^2 / \epsilon^2)$ which is exponentially small in the qubit count $N$ while increases polynomially with the circuit depth $D$. The curvature of local minima is concentrated to the QFI matrix times a fidelity-dependent constant which is positive at $p^2 > 1/d$. In practice, our results can be regarded as a quantum version of the no-free-lunch (NFL) theorem suggesting that no single QNN is universally the best-performing model for learning all target quantum states. We remark that compared to previous works, our findings first establish quantitative limits on good initial guesses and adaptive training methods for improving the learnability and scalability of QNNs.

In the technical part of our work, our ensemble arises from the unknown target state. Alternatively, if the QNN is sufficiently deep to form a subspace 2-design (cf. Appendix A.1) replacing the ensemble we used here, a different interpretation could be established with the same calculations: there are exponentially large proportion of local minima on some cross sections of the training landscape with a constant loss function value. However, it remains an open question what the scaling of the QNN depth is to constitute such a subspace 2-design, given that a local random quantum circuit of polynomial depth forms an approximate unitary $t$-design [76]. We would like to note that the case where the output and target states are mixed states is not covered due to the quantum nature of the hard-to-defining orthogonal ensemble of mixed states, which may be left for future research.

Future progress will necessitate more structured QNN architectures and optimization tools, where insights from the field of deep learning may prove beneficial. Our findings suggest that the unique characteristics and prior information of quantum systems must be thoughtfully encoded in the QNN in order to learn the state successfully, such as the low entanglement structure in the ground state [70], the local interactions in Hamiltonians [71, 77] and the adiabatic evolution from product states [72].

**Acknowledgement.** We would like to thank the helpful comments from the anonymous reviewers. Part of this work was done when H. Z., C. Z., M. J., and X. W. were at Baidu Research.

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

## Appendix for "Statistical Analysis of Quantum State Learning Process in Quantum Neural Networks"

## A  Preliminaries

### A.1  Subspace Haar integration

The central technique used in our work is the subspace Haar integration, i.e., a series of formulas on calculating Haar integrals over a certain subspace of the given Hilbert space. In this section, we give a brief introduction to the common Haar integrals and then the basic formulas on subspace Haar integrals used in our work together with the proofs.

Haar integrals refer to the matrix integrals over the $d$-degree unitary group $\mathcal{U}(d)$ with the Haar measure $d\mu$, which is the unique uniform measure on $\mathcal{U}(d)$ such that

$$\int_{\mathcal{U}(d)} d\mu(V)f(V) = \int_{\mathcal{U}(d)} d\mu(V)f(VU) = \int_{\mathcal{U}(d)} d\mu(V)f(UV), \tag{S1}$$

for any integrand $f$ and group element $U \in \mathcal{U}(d)$. If an ensemble $\mathbb{V}$ of unitaries $V$ matches the Haar measure up to the $t$-degree moment, i.e.,

$$\mathbb{E}_{V \in \mathbb{V}}[p_{t,t}(V)] = \int_{\mathcal{U}(d)} d\mu(V)p_{t,t}(V), \tag{S2}$$

then $\mathbb{V}$ is called a unitary $t$-design [78]. $p_{t,t}(V)$ denotes an arbitrary polynomial of degree at most $t$ in the entries of $V$ and at most $t$ in those of $V^\dagger$. $\mathbb{E}_{V \in \mathbb{V}}[\cdot]$ denotes the expectation over the ensemble $\mathbb{V}$. The Haar integrals over polynomials can be analytically solved and expressed into closed forms according to the following lemma.

**Lemma S1** *Let $\varphi : \mathcal{U}(d) \to \mathrm{GL}(\mathbb{C}^{d'})$ be an arbitrary representation of unitary group $\mathcal{U}(d)$. Suppose that the direct sum decomposition of $\varphi$ to irreducible representations is $\varphi = \bigoplus_{j,k} \phi_k^{(j)}$, where $\phi_k^{(j)}$ denotes the $k^{th}$ copy of the irreducible representation $\phi^{(j)}$. A set of orthonormal basis in the representation space of $\phi_k^{(j)}$ is denoted as $\{|v_{j,k,l}\rangle\}$. For an arbitrary linear operator $A : \mathbb{C}^{d'} \to \mathbb{C}^{d'}$, the following equality holds* [79]

$$\int_{\mathcal{U}(d)} \varphi(U)A\varphi(U)^\dagger d\mu(U) = \sum_{j,k,k'} \frac{\mathrm{tr}(Q_{j,k,k'}^\dagger A)}{\mathrm{tr}(Q_{j,k,k'}^\dagger Q_{j,k,k'})} Q_{j,k,k'}, \tag{S3}$$

*where $Q_{j,k,k'} = \sum_l |v_{j,k,l}\rangle\langle v_{j,k',l}|$ is the transfer operator from the representation subspace of $\phi_{k'}^{(j)}$ to that of $\phi_k^{(j)}$. The denominator on the right hand side of* (S3) *can be simplified as $\mathrm{tr}(Q_{j,k,k'}^\dagger Q_{j,k,k'}) = \mathrm{tr}(P_{j,k'}) = d_j$, where $P_{j,k} = \sum_l |v_{j,k,l}\rangle\langle v_{j,k,l}| = Q_{j,k,k}$ is the projector to the representation subspace of $\phi_k^{(j)}$ and $d_j$ is the dimension of the representation space of $\phi^{(j)}$.*

By choosing different representations of the unitary group $\mathcal{U}(d)$, some commonly used equalities can be derived, such as

$$\int_{\mathcal{U}(d)} VAV^\dagger d\mu(V) = \frac{\mathrm{tr}(A)}{d}I, \tag{S4}$$

$$\int_{\mathcal{U}(d)} V^\dagger AVBV^\dagger CV d\mu(V) = \frac{\mathrm{tr}(AC)\,\mathrm{tr}\,B}{d^2}I + \frac{d\,\mathrm{tr}\,A\,\mathrm{tr}\,C - \mathrm{tr}(AC)}{d(d^2-1)}\left(B - \frac{\mathrm{tr}\,B}{d}I\right), \tag{S5}$$

where $I$ is the identity operator on the $d$-dimensional Hilbert space $\mathcal{H}$. $A, B$ and $C$ are arbitrary linear operators on $\mathcal{H}$. According to the linearity of the integrals, the following equalities can be further derived

$$\int_{\mathcal{U}(d)} \text{tr}(VA)\,\text{tr}(V^\dagger B)d\mu(V) = \frac{\text{tr}(AB)}{d}, \tag{S6}$$

$$\int_{\mathcal{U}(d)} \text{tr}(V^\dagger AVB)\,\text{tr}(V^\dagger CVD)d\mu(V) = \frac{\text{tr}\,A\,\text{tr}\,B\,\text{tr}\,C\,\text{tr}\,D + \text{tr}(AC)\,\text{tr}(BD)}{d^2 - 1}$$
$$- \frac{\text{tr}(AC)\,\text{tr}\,B\,\text{tr}\,D + \text{tr}\,A\,\text{tr}\,C\,\text{tr}(BD)}{d(d^2 - 1)}, \tag{S7}$$

where $A, B, C$ and $D$ are arbitrary linear operators on $\mathcal{H}$.

The subspace Haar integration can be regarded as a simple generalization of the formulas above. Suppose that $\mathcal{H}_{\text{sub}}$ is a subspace with dimension $d_{\text{sub}}$ of the Hilbert space $\mathcal{H}$. $\mathbb{U}$ is an ensemble whose elements are unitaries in $\mathcal{H}$ with a block-diagonal structure $U = \bar{P} + PUP$. $P$ is the projector from $\mathcal{H}$ to $\mathcal{H}_{\text{sub}}$ and $PUP$ is a random unitary with the Haar measure on $\mathcal{H}_{\text{sub}}$. $\bar{P} = I - P$ is the projector from $\mathcal{H}$ to the orthogonal complement of $\mathcal{H}_{\text{sub}}$. Integrals with respect to such an ensemble $\mathbb{U}$ are dubbed as "*subspace Haar integrals*", which can be reduced back to the common Haar integrals by taking $\mathcal{H}_{\text{sub}} = \mathcal{H}$. The corresponding formulas of subspace Haar integrals are developed in the following lemmas, where $\mathbb{E}_{U \in \mathbb{U}}[\cdot] = \mathbb{E}_{\mathbb{U}}[\cdot]$ denotes the expectation with respect to the ensemble $\mathbb{U}$.

**Lemma S2** *The expectation of a single element $U \in \mathbb{U}$ with respect to the ensemble $\mathbb{U}$ equals to the projector to the orthogonal complement, i.e.,*

$$\mathbb{E}_{\mathbb{U}}[U] = I - P. \tag{S8}$$

**Proof** The fact that Haar integrals of inhomogenous polynomials $p_{t,t'}$ with $t \neq t'$ over the whole space equals to zero leads to the vanishment of the block in $\mathcal{H}_{\text{sub}}$, i.e.,

$$\mathbb{E}_{\mathbb{U}}[U] = \mathbb{E}_{\mathbb{U}}\left[\bar{P} + PUP\right] = \bar{P}, \tag{S9}$$

which is just the projector to the orthogonal complement $\mathcal{H}_{\text{sub}}$. ∎

Similarly, we know all the subspace Haar integrals involving only $U$ or $U^\dagger$ will leave a projector after integration. For example, it holds that $\mathbb{E}_{\mathbb{U}}[UAU] = \bar{P}A\bar{P}$ for an arbitrary linear operator $A$.

**Lemma S3** *For an arbitrary linear operator $A$ on $\mathcal{H}$, the expectation of $U^\dagger AU$ with respect to the random variable $U \in \mathbb{U}$ is*

$$\mathbb{E}_{\mathbb{U}}\left[U^\dagger AU\right] = \frac{\text{tr}(PA)}{d_{\text{sub}}}P + (I - P)A(I - P). \tag{S10}$$

**Proof** Eq. (S10) can be seen as a special case of Lemma S1 since $U$ can be seen as the complete reducible representation of $\mathcal{U}(d_{\text{sub}})$ composed of $(d - d_{\text{sub}})$ trivial representations $\phi_k^{(1)}$ with $k = 1, ..., (d - d_{\text{sub}})$ and one natural representation $\phi^{(2)}$. This gives rise to

$$\sum_{k,k'} \frac{\text{tr}(Q_{1,k,k'}^\dagger A)}{\text{tr}(Q_{1,k,k'}^\dagger Q_{1,k,k'})} Q_{1,k,k'} = \bar{P}A\bar{P},$$
$$\frac{\text{tr}(Q_2^\dagger A)}{\text{tr}(Q_2^\dagger Q_2)} Q_2 = \frac{\text{tr}(PA)}{d_{\text{sub}}} P. \tag{S11}$$

Alternatively, Eq. (S10) can just be seen as a result of the block matrix multiplication, i.e.,

$$\mathbb{E}_{\mathbb{U}}[U^\dagger AU] = \mathbb{E}_{\mathbb{U}}[(\bar{P} + PU^\dagger P)A(\bar{P} + PUP)]$$
$$= \mathbb{E}_{\mathbb{U}}[\bar{P}A\bar{P} + \bar{P}APUP + PU^\dagger PA\bar{P} + PU^\dagger PAPUP]$$
$$= \bar{P}A\bar{P} + \frac{\text{tr}(PAP)}{d_{\text{sub}}}P, \tag{S12}$$

where $\text{tr}(PAP) = \text{tr}(P^2A) = \text{tr}(PA)$. ∎

**Corollary S4** *Suppose $|\varphi\rangle$ is a Haar-random pure state in $\mathcal{H}_{\mathrm{sub}}$. For arbitrary linear operators $A$ on $\mathcal{H}$, the following equality holds*

$$\mathbb{E}_{\varphi}\left[\langle\varphi|A|\varphi\rangle\right] = \frac{\mathrm{tr}(PA)}{d_{\mathrm{sub}}}, \tag{S13}$$

*where $\mathbb{E}_{\varphi}[\cdot]$ is the expectation with respect to the random state $|\varphi\rangle$.*

**Proof** Suppose $|\varphi_0\rangle$ is an arbitrary fixed state in $\mathcal{H}_{\mathrm{sub}}$. The random state $|\varphi\rangle$ can be written in terms of $U \in \mathbb{U}$ as $|\varphi\rangle = U|\varphi_0\rangle$ such that

$$\mathbb{E}_{\varphi}\left[\langle\varphi|A|\varphi\rangle\right] = \mathbb{E}_{U\in\mathbb{U}}\left[\langle\varphi_0|U^{\dagger}AU|\varphi_0\rangle\right]. \tag{S14}$$

Eq. (S13) is naturally obtained from Lemma S3 by taking the expectation over $|\varphi_0\rangle$ which satisfies $P|\varphi_0\rangle = |\varphi_0\rangle$ and $\bar{P}|\varphi_0\rangle = 0$. ∎

**Lemma S5** *For arbitrary linear operators $A, B, C$ on $\mathcal{H}$ and $U \in \mathbb{U}$, the following equality holds*

$$
\begin{aligned}
&\mathbb{E}_{\mathbb{U}}\left[U^{\dagger}AUBU^{\dagger}CU\right] \\
&= \bar{P}A\bar{P}B\bar{P}C\bar{P} + \frac{\mathrm{tr}\left(PB\right)}{d_{\mathrm{sub}}}\bar{P}APC\bar{P} + \frac{\mathrm{tr}\left(PC\right)}{d_{\mathrm{sub}}}\bar{P}A\bar{P}BP + \frac{\mathrm{tr}\left(PA\right)}{d_{\mathrm{sub}}}PB\bar{P}C\bar{P} \\
&\quad + \frac{\mathrm{tr}\left(PA\bar{P}B\bar{P}C\right)}{d_{\mathrm{sub}}}P + \frac{\mathrm{tr}(PAPC)\,\mathrm{tr}(PB)}{d_{\mathrm{sub}}^2}P \\
&\quad + \frac{d_{\mathrm{sub}}\,\mathrm{tr}(PA)\,\mathrm{tr}(PC) - \mathrm{tr}(PAPC)}{d_{\mathrm{sub}}(d_{\mathrm{sub}}^2 - 1)}\left(PBP - \frac{\mathrm{tr}(PB)}{d_{\mathrm{sub}}}P\right).
\end{aligned}
\tag{S15}
$$

**Proof** Here we simply employ the block matrix multiplication to prove this equality. We denote the $2 \times 2$ blocks with indices $\begin{pmatrix} 11 & 12 \\ 21 & 22 \end{pmatrix}$ respectively where the index 2 corresponds to $\mathcal{H}_{\mathrm{sub}}$. Thus the random unitary $U$ can be written as $U = \begin{pmatrix} I_{11} & 0 \\ 0 & U_{22} \end{pmatrix}$ where $I_{11}$ is the identity matrix on the orthogonal complement of $\mathcal{H}_{\mathrm{sub}}$ and $U_{22}$ is a Haar-random unitary on $\mathcal{H}_{\mathrm{sub}}$. The integrand becomes

$$U^{\dagger}AUBU^{\dagger}CU = \begin{pmatrix} A_{11} & A_{12}U_{22} \\ U_{22}^{\dagger}A_{21} & U_{22}^{\dagger}A_{22}U_{22} \end{pmatrix}\begin{pmatrix} B_{11} & B_{12} \\ B_{21} & B_{22} \end{pmatrix}\begin{pmatrix} C_{11} & C_{12}U_{22} \\ U_{22}^{\dagger}C_{21} & U_{22}^{\dagger}C_{22}U_{22} \end{pmatrix}. \tag{S16}$$

The four matrix elements of the multiplication results are

$$
\begin{aligned}
11:&\ A_{11}B_{11}C_{11} + A_{12}U_{22}B_{21}C_{11} + A_{11}B_{12}U_{22}^{\dagger}C_{11} + A_{12}U_{22}B_{22}U_{22}^{\dagger}C_{21}, \\
12:&\ A_{11}B_{11}C_{12}U_{22} + A_{12}U_{22}B_{21}C_{12}U_{22} + A_{11}B_{12}U_{22}^{\dagger}C_{22}U_{22} + A_{12}U_{22}B_{22}U_{22}^{\dagger}C_{22}U_{22}, \\
21:&\ U_{22}^{\dagger}A_{21}B_{11}C_{11} + U_{22}^{\dagger}A_{22}U_{22}B_{21}C_{11} + U_{22}^{\dagger}A_{21}B_{12}U_{22}^{\dagger}C_{21} + U_{22}^{\dagger}A_{22}U_{22}B_{22}U_{22}^{\dagger}C_{21}, \\
22:&\ U_{22}^{\dagger}A_{21}B_{11}C_{12}U_{22} + U_{22}^{\dagger}A_{22}U_{22}B_{21}C_{12}U_{22} + U_{22}^{\dagger}A_{21}B_{12}U_{22}^{\dagger}C_{22}U_{22} \\
&\ + U_{22}^{\dagger}A_{22}U_{22}B_{22}U_{22}^{\dagger}C_{22}U_{22}.
\end{aligned}
\tag{S17}
$$

Since inhomogeneous Haar integrals always vanish on $\mathcal{H}_{\mathrm{sub}}$, the elements above can be reduced to

$$
\begin{aligned}
11:&\ A_{11}B_{11}C_{11} + A_{12}U_{22}B_{22}U_{22}^{\dagger}C_{21}, \\
12:&\ A_{11}B_{12}U_{22}^{\dagger}C_{22}U_{22}, \quad 21:\ U_{22}^{\dagger}A_{22}U_{22}B_{21}C_{11}, \\
22:&\ U_{22}^{\dagger}A_{21}B_{11}C_{12}U_{22} + U_{22}^{\dagger}A_{22}U_{22}B_{22}U_{22}^{\dagger}C_{22}U_{22}.
\end{aligned}
\tag{S18}
$$

Let $d_2 = d_{\text{sub}} = \dim \mathcal{H}_{\text{sub}}$ and $I_{22}$ be the identity matrix in $\mathcal{H}_{\text{sub}}$. Utilizing Eqs. (S4) and (S5), the expectation of each block becomes

$$
\begin{aligned}
11 : \; & A_{11}B_{11}C_{11} + \frac{\operatorname{tr} B_{22}}{d_2} A_{12}C_{21}, \\[4pt]
12 : \; & \frac{\operatorname{tr} C_{22}}{d_2} A_{11}B_{12}, \quad 21 : \; \frac{\operatorname{tr} A_{22}}{d_2} B_{21}C_{11}, \\[4pt]
22 : \; & \frac{\operatorname{tr}(A_{21}B_{11}C_{12})}{d_2} I_{22} + \frac{\operatorname{tr}(A_{22}C_{22})\operatorname{tr}(B_{22})}{d_2^2} I_{22} \\[4pt]
& + \frac{d_2 \operatorname{tr}(A_{22})\operatorname{tr}(C_{22}) - \operatorname{tr}(A_{22}C_{22})}{d_2(d_2^2 - 1)} \left( B_{22} - \frac{\operatorname{tr}(B_{22})}{d_2} I_{22} \right).
\end{aligned}
\tag{S19}
$$

Written in terms of subspace projectors $P$ and $\bar{P}$, the results become exactly as Eq. (S15). ∎

**Corollary S6** *Suppose $|\varphi\rangle$ is a Haar-random pure state in $\mathcal{H}_{\text{sub}}$. For arbitrary linear operators $A$ on $\mathcal{H}$, the following equality holds*

$$
\mathbb{E}_\varphi \left[ \langle\varphi|A|\varphi\rangle^2 \right] = \frac{\operatorname{tr}((PA)^2) + (\operatorname{tr}(PA))^2}{d_{\text{sub}}(d_{\text{sub}} + 1)},
\tag{S20}
$$

*where $\mathbb{E}_\varphi[\cdot]$ is the expectation with respect to the random state $|\varphi\rangle$.*

**Proof** Suppose $|\varphi_0\rangle$ is an arbitrary fixed state in $\mathcal{H}_{\text{sub}}$. The random state $|\varphi\rangle$ can be written in terms of $U \in \mathbb{U}$ as $|\varphi\rangle = U|\varphi_0\rangle$ such that

$$
\mathbb{E}_\varphi \left[ (\langle\varphi|A|\varphi\rangle)^2 \right] = \mathbb{E}_{U \in \mathbb{U}} \left[ \langle\varphi_0|U^\dagger A U|\varphi_0\rangle\langle\varphi_0|U^\dagger A U|\varphi_0\rangle \right].
\tag{S21}
$$

Eq. (S20) is naturally obtained from Lemma S5 by taking $C = A$ and $B = |\varphi_0\rangle\langle\varphi_0|$ which satisfies $\bar{P}B = B\bar{P} = 0$, $PBP = B$ and $\operatorname{tr} B = 1$. ∎

**Lemma S7** *For arbitrary linear operators $A, B, C, D$ on $\mathcal{H}$ and $U \in \mathbb{U}$, the following equality holds*

$$
\begin{aligned}
\mathbb{E}_{\mathbb{U}} & \left[ \operatorname{tr}(U^\dagger A U B) \operatorname{tr}(U^\dagger C U D) \right] = \operatorname{tr}(\bar{P}A\bar{P}B)\operatorname{tr}(\bar{P}C\bar{P}D) \\[4pt]
& + \frac{\operatorname{tr}(\bar{P}A\bar{P}B)\operatorname{tr}(PC)\operatorname{tr}(PD)}{d_{\text{sub}}} + \frac{\operatorname{tr}(\bar{P}C\bar{P}D)\operatorname{tr}(PA)\operatorname{tr}(PB)}{d_{\text{sub}}} \\[4pt]
& + \frac{\operatorname{tr}(PB\bar{P}APC\bar{P}D)}{d_{\text{sub}}} + \frac{\operatorname{tr}(PA\bar{P}BPD\bar{P}C)}{d_{\text{sub}}} \\[4pt]
& + \frac{\operatorname{tr}(PA)\operatorname{tr}(PB)\operatorname{tr}(PC)\operatorname{tr}(PD) + \operatorname{tr}(PAPC)\operatorname{tr}(PBPD)}{d_{\text{sub}}^2 - 1} \\[4pt]
& - \frac{\operatorname{tr}(PAPC)\operatorname{tr}(PB)\operatorname{tr}(PD) + \operatorname{tr}(PA)\operatorname{tr}(PC)\operatorname{tr}(PBPD)}{d_{\text{sub}}(d_{\text{sub}}^2 - 1)}.
\end{aligned}
\tag{S22}
$$

**Proof** Similarly with the proof of Lemma S5, the block matrix multiplication gives

$$
\begin{aligned}
\operatorname{tr}(U^\dagger A U B) &= \operatorname{tr}\left[ \begin{pmatrix} A_{11} & A_{12}U_{22} \\ U_{22}^\dagger A_{21} & U_{22}^\dagger A_{22}U_{22} \end{pmatrix} \begin{pmatrix} B_{11} & B_{12} \\ B_{21} & B_{22} \end{pmatrix} \right] \\[4pt]
&= \operatorname{tr}(A_{11}B_{11}) + \operatorname{tr}(A_{12}U_{22}B_{21}) + \operatorname{tr}(U_{22}^\dagger A_{21}B_{12}) + \operatorname{tr}(U_{22}^\dagger A_{22}U_{22}B_{22}).
\end{aligned}
\tag{S23}
$$

Hence we have

$$
\begin{aligned}
\mathbb{E}_{\mathbb{U}} & \left[ \operatorname{tr}(U^\dagger A U B) \operatorname{tr}(U^\dagger C U D) \right] = \mathbb{E}_{\mathbb{U}} \big[ \operatorname{tr}(A_{11}B_{11})\operatorname{tr}(C_{11}D_{11}) \\[4pt]
& + \operatorname{tr}(A_{11}B_{11})\operatorname{tr}(U_{22}^\dagger C_{22}U_{22}D_{22}) + \operatorname{tr}(C_{11}D_{11})\operatorname{tr}(U_{22}^\dagger A_{22}U_{22}B_{22}) \\[4pt]
& + \operatorname{tr}(A_{12}U_{22}B_{21})\operatorname{tr}(U_{22}^\dagger C_{21}D_{12}) + \operatorname{tr}(U_{22}^\dagger A_{21}B_{12})\operatorname{tr}(C_{12}U_{22}D_{21}) \\[4pt]
& + \operatorname{tr}(U_{22}^\dagger A_{22}U_{22}B_{22})\operatorname{tr}(U_{22}^\dagger C_{22}U_{22}D_{22}) \big].
\end{aligned}
\tag{S24}
$$

where all inhomogeneous terms have been ignored. Utilizing Eqs. (S4), (S6) and (S7), the expectation becomes

$$
\begin{aligned}
\mathbb{E}_{\mathbb{U}}\left[\operatorname{tr}(U^{\dagger}AUB)\operatorname{tr}(U^{\dagger}CUD)\right] &= \operatorname{tr}(A_{11}B_{11})\operatorname{tr}(C_{11}D_{11}) \\
&+ \frac{\operatorname{tr}(A_{11}B_{11})\operatorname{tr}(C_{22})\operatorname{tr}(D_{22})}{d_2} + \frac{\operatorname{tr}(C_{11}D_{11})\operatorname{tr}(A_{22})\operatorname{tr}(B_{22})}{d_2} \\
&+ \frac{\operatorname{tr}(B_{21}A_{12}C_{21}D_{12})}{d_2} + \frac{\operatorname{tr}(A_{21}B_{12}D_{21}C_{12})}{d_2} \\
&+ \frac{1}{d_2^2-1}(\operatorname{tr}(A_{22})\operatorname{tr}(B_{22})\operatorname{tr}(C_{22})\operatorname{tr}(D_{22}) + \operatorname{tr}(A_{22}C_{22})\operatorname{tr}(B_{22}D_{22})) \\
&- \frac{1}{d_2(d_2^2-1)}(\operatorname{tr}(A_{22}C_{22})\operatorname{tr}(B_{22})\operatorname{tr}(D_{22}) + \operatorname{tr}(A_{22})\operatorname{tr}(C_{22})\operatorname{tr}(B_{22}D_{22}))
\end{aligned}
\tag{S25}
$$

Written in terms of subspace projectors $P$ and $\bar{P}$, the results become exactly as Eq. (S22). ∎

Finally, similar to the unitary $t$-design, we introduce the concept of "subspace $t$-design". If an ensemble $\mathbb{W}$ of unitaries $V$ matches the ensemble $\mathbb{U}$ rotating the subspace $\mathcal{H}_{\text{sub}}$ up to the $t$-degree moment, then $\mathbb{W}$ is called a subspace unitary $t$-design with respect to $\mathcal{H}_{\text{sub}}$. In the main text, the ensemble comes from the unknown target state. Alternatively, if a random QNN $\mathbf{U}(\boldsymbol{\theta})$ with some constraints such as keeping the loss function constant $\mathcal{L}(\boldsymbol{\theta}) = \mathcal{L}_0$, i.e.,

$$
\mathbb{W} = \mathbf{U}(\Theta), \quad \Theta = \{\boldsymbol{\theta} \mid \mathcal{L}(\boldsymbol{\theta}) = \mathcal{L}_0\},
\tag{S26}
$$

forms a approximate subspace 2-design, then similar results as in the main text can be established yet with a different interpretation: there is an exponentially large proportion of local minima on a constant-loss-section of the training landscape.

## A.2 Perturbation on positive definite matrices

To identify whether a parameter point is a local minimum, we need to check whether the Hessian matrix is positive definite, where the following sufficient condition is used in the proof of our main theorem in the next section.

**Lemma S8** *Suppose $X$ is a positive definite matrix and $Y$ is a Hermitian matrix. If the distance between $Y$ and $X$ is smaller than the minimal eigenvalue of $X$, i.e., $\|Y - X\|_{\infty} < \|X^{-1}\|_{\infty}^{-1}$, then $Y$ is positive definite. Here $\|\cdot\|_{\infty}$ denotes the Schatten-$\infty$ norm.*

**Proof** For an arbitrary vector $|v\rangle$, we have

$$
\langle v|Y|v\rangle = \langle v|X|v\rangle + \langle v|Y-X|v\rangle \geq \|X^{-1}\|_{\infty}^{-1} - \|Y-X\|_{\infty} > 0.
\tag{S27}
$$

Note that $\|X^{-1}\|_{\infty}^{-1}$ just represents the minimal eigenvalue of the positive matrix $X$. Thus, $Y$ is positive definite. ∎

## A.3 Tail inequalities

In order to bound the probability of avoiding local minima, we need to use some "tail inequalities" in probability theory, especially the generalized Chebyshev's inequality for matrices, which we summarize below for clarity.

**Lemma S9** (Markov's inequality) *For a non-negative random variable $X$ and $a > 0$, the probability that $X$ is at least $a$ is upper bounded by the expectation of $X$ divided by $a$, i.e.,*

$$
\Pr[X \geq a] \leq \frac{\mathbb{E}[X]}{a}.
\tag{S28}
$$

**Proof** The expectation can be rewritten and bounded as

$$
\begin{aligned}
\mathbb{E}[X] &= \Pr[X < a] \cdot \mathbb{E}[X \mid X < a] + \Pr[X \geq a] \cdot \mathbb{E}[X \mid X \geq a] \\
&\geq \Pr[X \geq a] \cdot \mathbb{E}[X \mid X \geq a] \geq \Pr[X \geq a] \cdot a.
\end{aligned}
\tag{S29}
$$

Thus we have $\Pr[X \geq a] \leq \mathbb{E}[X]/a$. ∎

**Lemma S10** (Chebyshev's inequality) *For a real random variable $X$ and $\varepsilon > 0$, the probability that $X$ deviates from the expectation $\mathbb{E}[X]$ by $\varepsilon$ is upper bounded by the variance of $X$ divided by $\varepsilon^2$, i.e.,*

$$\Pr[|X - \mathbb{E}[X]| \geq \varepsilon] \leq \frac{\mathrm{Var}[X]}{\varepsilon^2}. \tag{S30}$$

**Proof** Applying Markov's inequality in Lemma S9 to the random variable $(X - \mathbb{E}[X])^2$ gives

$$\Pr[|X - \mathbb{E}[X]| \geq \varepsilon] = \Pr[(X - \mathbb{E}[X])^2 \geq \varepsilon^2] \leq \frac{\mathbb{E}[(X - \mathbb{E}[X])^2]}{\varepsilon^2} = \frac{\mathrm{Var}[X]}{\varepsilon^2}. \tag{S31}$$

Alternatively, the proof can be carried out similarly as in Eq. (S29) with respect to $(X - \mathbb{E}[X])^2$. ∎

**Lemma S11** (Chebyshev's inequality for matrices) *For a random matrix $X$ and $\varepsilon > 0$, the probability that $X$ deviates from the expectation $\mathbb{E}[X]$ by $\varepsilon$ in terms of the norm $\|\cdot\|_\alpha$ satisfies*

$$\Pr\left[\|X - \mathbb{E}[X]\|_\alpha \geq \varepsilon\right] \leq \frac{\sigma_\alpha^2}{\varepsilon^2} \tag{S32}$$

*where $\sigma_\alpha^2 = \mathbb{E}[\|X - \mathbb{E}[X]\|_\alpha^2]$ denotes the variance of $X$ in terms of the norm $\|\cdot\|_\alpha$.*

**Proof** Applying Markov's inequality in Lemma S9 to the random variable $\|X - \mathbb{E}[X]\|_\alpha^2$ gives

$$\Pr[\|X - \mathbb{E}[X]\|_\alpha \geq \varepsilon] = \Pr[\|X - \mathbb{E}[X]\|_\alpha^2 \geq \varepsilon^2] \leq \frac{\mathbb{E}[\|X - \mathbb{E}[X]\|_\alpha^2]}{\varepsilon^2} = \frac{\sigma_\alpha^2}{\varepsilon^2}. \tag{S33}$$

Note that here the expectation $\mathbb{E}[X]$ is still a matrix while the "variance" $\sigma_\alpha^2$ is a real number. ∎

### A.4 Quantum Fisher information matrix

Given a parameterized pure quantum state $|\psi(\boldsymbol{\theta})\rangle$, the quantum Fisher information (QFI) matrix $\mathcal{F}_{\mu\nu}$ [64] is defined as the Riemannian metric induced from the Bures fidelity distance $d_\mathrm{f}(\boldsymbol{\theta}, \boldsymbol{\theta}') = 1 - |\langle\psi(\boldsymbol{\theta})|\psi(\boldsymbol{\theta}')\rangle|^2$ (up to a factor 2 depending on convention), i.e.,

$$\mathcal{F}_{\mu\nu}(\boldsymbol{\theta}) = \left.\frac{\partial^2}{\partial\delta_\mu\partial\delta_\nu}d_\mathrm{f}(\boldsymbol{\theta}, \boldsymbol{\theta} + \boldsymbol{\delta})\right|_{\boldsymbol{\delta}=0} = -2\,\mathrm{Re}\left[\langle\partial_\mu\partial_\nu\psi|\psi\rangle + \langle\partial_\mu\psi|\psi\rangle\langle\psi|\partial_\nu\psi\rangle\right]. \tag{S34}$$

Note that $|\partial_\mu\psi\rangle$ actually refers to $\frac{\partial}{\partial\theta_\mu}|\psi(\boldsymbol{\theta})\rangle$. Using the normalization condition

$$\begin{aligned}
&\langle\psi|\psi\rangle = 1, \\
&\partial_\mu(\langle\psi|\psi\rangle) = \langle\partial_\mu\psi|\psi\rangle + \langle\psi|\partial_\mu\psi\rangle = 2\,\mathrm{Re}\left[\langle\partial_\mu\psi|\psi\rangle\right] = 0, \\
&\partial_\mu\partial_\nu(\langle\psi|\psi\rangle) = 2\,\mathrm{Re}\left[\langle\partial_\mu\partial_\nu\psi|\psi\rangle + \langle\partial_\mu\psi|\partial_\nu\psi\rangle\right] = 0,
\end{aligned} \tag{S35}$$

the QFI can be rewritten as

$$\mathcal{F}_{\mu\nu} = 2\,\mathrm{Re}\left[\langle\partial_\mu\psi|\partial_\nu\psi\rangle - \langle\partial_\mu\psi|\psi\rangle\langle\psi|\partial_\nu\psi\rangle\right]. \tag{S36}$$

The QFI characterizes the sensibility of a parameterized quantum state to a small change of parameters, and can be viewed as the real part of the quantum geometric tensor.

## B Detailed proofs

In this section, we provide the detailed proofs of Lemma 1, Theorem 2 and Proposition 3 in the main text. Here we use $d$ to denote the dimension of the Hilbert space. For a qubit system with $N$ qubits, we have $d = 2^N$. As in the main text, we represent the value of a certain function at $\boldsymbol{\theta} = \boldsymbol{\theta}^*$ by appending the superscript "$*$" for simplicity of notation, e.g., $\nabla\mathcal{L}|_{\boldsymbol{\theta}=\boldsymbol{\theta}^*}$ as $\nabla\mathcal{L}^*$ and $H_\mathcal{L}|_{\boldsymbol{\theta}=\boldsymbol{\theta}^*}$ as $H_\mathcal{L}^*$. In addition, for a parameterized quantum circuit $\mathbf{U}(\boldsymbol{\theta}) = \prod_{\mu=1}^M U_\mu(\theta_\mu)W_\mu$, we introduce the notation $V_{\alpha\to\beta} = \prod_{\mu=\alpha}^\beta U_\mu W_\mu$ if $\alpha \leq \beta$ and $V_{\alpha\to\beta} = I$ if $\alpha > \beta$. Note that the product $\prod_\mu$ is by default in the increasing order from the right to the left. The derivative with respect to the parameter $\theta_\mu$ is simply denoted as $\partial_\mu = \frac{\partial}{\partial\theta_\mu}$. We remark that our results hold for all kinds of input states into QNNs in spite that we use $|0\rangle^{\otimes N}$ in the definition of $|\psi(\boldsymbol{\theta})\rangle$ for simplicity.

**Lemma 1** *The expectation and variance of the gradient $\nabla\mathcal{L}$ and Hessian matrix $H_\mathcal{L}$ of the fidelity loss function $\mathcal{L}(\boldsymbol{\theta}) = 1 - |\langle\phi|\psi(\boldsymbol{\theta})\rangle|^2$ at $\boldsymbol{\theta} = \boldsymbol{\theta}^*$ with respect to the target state ensemble $\mathbb{T}$ satisfy*

$$\mathbb{E}_\mathbb{T}\left[\nabla\mathcal{L}^*\right] = 0, \quad \mathrm{Var}_\mathbb{T}[\partial_\mu\mathcal{L}^*] = f_1(p,d)\mathcal{F}^*_{\mu\mu}. \tag{S37}$$

$$\mathbb{E}_\mathbb{T}\left[H^*_\mathcal{L}\right] = \frac{dp^2-1}{d-1}\mathcal{F}^*, \quad \mathrm{Var}_\mathbb{T}[\partial_\mu\partial_\nu\mathcal{L}^*] \leq f_2(p,d)\|\Omega_\mu\|^2_\infty\|\Omega_\nu\|^2_\infty. \tag{S38}$$

*where $\mathcal{F}$ denote the QFI matrix. $f_1$ and $f_2$ are functions of the overlap $p$ and the Hilbert space dimension $d$, i.e.,*

$$f_1(p,d) = \frac{p^2(1-p^2)}{d-1}, \quad f_2(p,d) = \frac{32(1-p^2)}{d-1}\left[p^2 + \frac{2(1-p^2)}{d}\right]. \tag{S39}$$

**Proof** Using the decomposition in Eq. (4), the loss function can be expressed by

$$\mathcal{L} = 1 - \langle\phi|\varrho|\phi\rangle = 1 - p^2\langle\psi^*|\varrho|\psi^*\rangle - (1-p^2)\langle\psi^\perp|\varrho|\psi^\perp\rangle - 2p\sqrt{1-p^2}\,\mathrm{Re}\left(\langle\psi^\perp|\varrho|\psi^*\rangle\right), \tag{S40}$$

where $\varrho(\boldsymbol{\theta}) = |\psi(\boldsymbol{\theta})\rangle\langle\psi(\boldsymbol{\theta})|$ denotes the density matrix of the output state from the QNN. According to Lemma S2 and Corollary S4, the expectation of the loss function with respect to the ensemble $\mathbb{T}$ can be calculated as

$$\begin{aligned}
\mathbb{E}_\mathbb{T}\left[\mathcal{L}(\boldsymbol{\theta})\right] &= 1 - p^2\langle\psi^*|\varrho|\psi^*\rangle - (1-p^2)\frac{\mathrm{tr}[(I - |\psi^*\rangle\langle\psi^*|)\varrho]}{d-1} \\
&= 1 - p^2 + \frac{dp^2-1}{d-1}g(\boldsymbol{\theta}),
\end{aligned} \tag{S41}$$

where $g(\boldsymbol{\theta}) = 1 - \langle\psi^*|\varrho(\boldsymbol{\theta})|\psi^*\rangle$ denotes the fidelity distance between the output states at $\boldsymbol{\theta}$ and $\boldsymbol{\theta}^*$. By definition, $g(\boldsymbol{\theta})$ takes the global minimum at $\boldsymbol{\theta} = \boldsymbol{\theta}^*$, i.e., at $\varrho = |\psi^*\rangle\langle\psi^*|$. Thus the commutation between the expectation and differentiation gives

$$\begin{aligned}
\mathbb{E}_\mathbb{T}\left[\nabla\mathcal{L}^*\right] &= \nabla\left(\mathbb{E}_\mathbb{T}\left[C\right]\right)|_{\boldsymbol{\theta}=\boldsymbol{\theta}^*} = \frac{dp^2-1}{d-1}\nabla g(\boldsymbol{\theta})|_{\boldsymbol{\theta}=\boldsymbol{\theta}^*} = 0, \\
\mathbb{E}_\mathbb{T}\left[H^*_\mathcal{L}\right] &= \frac{dp^2-1}{d-1}H_g(\boldsymbol{\theta})|_{\boldsymbol{\theta}=\boldsymbol{\theta}^*} = \frac{dp^2-1}{d-1}H_g(\boldsymbol{\theta})|_{\boldsymbol{\theta}=\boldsymbol{\theta}^*} = \frac{dp^2-1}{d-1}\mathcal{F}^*.
\end{aligned} \tag{S42}$$

Note that $H_g(\boldsymbol{\theta})|_{\boldsymbol{\theta}=\boldsymbol{\theta}^*}$ is actually the QFI matrix $\mathcal{F}$ of $|\psi(\boldsymbol{\theta})\rangle$ at $\boldsymbol{\theta} = \boldsymbol{\theta}^*$ (see Appendix A.4), which is always positive semidefinite. To estimate the variance, we need to calculate the expression of derivatives first due to the non-linearity of the variance, unlike the case of Eq. (S42) where the operations of taking the expectation and derivative is exchanged. The first order derivative can be expressed by

$$\partial_\mu\mathcal{L} = -\langle\phi|D_\mu|\phi\rangle = -p^2\langle\psi^*|D_\mu|\psi^*\rangle - q^2\langle\psi^\perp|D_\mu|\psi^\perp\rangle - 2pq\,\mathrm{Re}\left(\langle\psi^\perp|D_\mu|\psi^*\rangle\right). \tag{S43}$$

where $q = \sqrt{1-p^2}$ and $D_\mu = \partial_\mu\varrho$ is a traceless Hermitian operator since $\mathrm{tr}\,D_\mu = \partial_\mu(\mathrm{tr}\,\varrho) = 0$. At $\boldsymbol{\theta} = \boldsymbol{\theta}^*$, the operator $D_\mu$ is reduced to $D^*_\mu$ which satisfies several useful properties

$$\begin{aligned}
D^*_\mu &= [\partial_\mu(|\psi\rangle\langle\psi|)]^* = |\partial_\mu\psi^*\rangle\langle\psi^*| + |\psi^*\rangle\langle\partial_\mu\psi^*|, \\
\langle\psi^*|D^*_\mu|\psi^*\rangle &= \langle\psi^*|\partial_\mu\psi^*\rangle + \langle\partial_\mu\psi^*|\psi^*\rangle = \partial_\mu(\langle\psi|\psi\rangle)^* = 0, \\
\langle\psi^\perp|D^*_\mu|\psi^\perp\rangle &= \langle\psi^\perp|\partial_\mu\psi^*\rangle\langle\psi^*|\psi^\perp\rangle + \langle\psi^\perp|\psi^*\rangle\langle\partial_\mu\psi^*|\psi^\perp\rangle = 0, \\
\langle\psi^\perp|D^*_\mu|\psi^*\rangle &= \langle\psi^\perp|\partial_\mu\psi^*\rangle\langle\psi^*|\psi^*\rangle + \langle\psi^\perp|\psi^*\rangle\langle\partial_\mu\psi^*|\psi^*\rangle = \langle\psi^\perp|\partial_\mu\psi^*\rangle,
\end{aligned} \tag{S44}$$

where we have used the facts of $\langle\psi|\psi\rangle = 1$ and $\langle\psi^*|\psi^\perp\rangle = 0$. Note that $|\partial_\mu\psi^*\rangle$ actually refers to $(\partial_\mu|\psi\rangle)^*$. Thus the variance of the first order derivative at $\boldsymbol{\theta} = \boldsymbol{\theta}^*$ becomes

$$\begin{aligned}
\mathrm{Var}_\mathbb{T}[\partial_\mu\mathcal{L}^*] &= \mathbb{E}_\mathbb{T}\left[(\partial_\mu\mathcal{L}^* - \mathbb{E}_\mathbb{T}[\partial_\mu\mathcal{L}^*])^2\right] = \mathbb{E}_\mathbb{T}\left[(\partial_\mu\mathcal{L}^*)^2\right] \\
&= 4p^2q^2\,\mathbb{E}_\mathbb{T}\left[\left(\mathrm{Re}\langle\psi^\perp|\partial_\mu\psi^*\rangle\right)^2\right].
\end{aligned} \tag{S45}$$

According to Lemma S2 and Corollary S4, it holds that

$$\begin{aligned}
\mathbb{E}_\mathbb{T}\left[\left(\mathrm{Re}\langle\psi^\perp|\partial_\mu\psi^*\rangle\right)^2\right] &= \frac{1}{2}\mathbb{E}_\mathbb{T}\left[\langle\psi^\perp|\partial_\mu\psi^*\rangle\langle\partial_\mu\psi^*|\psi^\perp\rangle\right] \\
&= \frac{\langle\partial_\mu\psi^*|\partial_\mu\psi^*\rangle - \langle\psi^*|\partial_\mu\psi^*\rangle\langle\partial_\mu\psi^*|\psi^*\rangle}{2(d-1)} = \frac{\mathcal{F}^*_{\mu\mu}}{4(d-1)},
\end{aligned} \tag{S46}$$

where $\mathcal{F}_{\mu\mu}$ is the QFI diagonal element. Using the generators in the PQC, $\mathcal{F}_{\mu\mu}$ could be expressed as

$$\mathcal{F}_{\mu\mu} = 2\left(\langle\psi|\tilde{\Omega}_\mu^2|\psi\rangle - \langle\psi|\tilde{\Omega}_\mu|\psi\rangle^2\right), \tag{S47}$$

where $\tilde{\Omega}_\mu = V_{\mu\to M}\Omega_\mu V_{\mu\to M}^\dagger$. Finally, the variance of $\partial_\mu\mathcal{L}$ at $\boldsymbol{\theta} = \boldsymbol{\theta}^*$ equals to

$$\mathrm{Var}_\mathbb{T}[\partial_\mu\mathcal{L}^*] = 4p^2q^2 \cdot \frac{\mathcal{F}_{\mu\mu}^*}{4(d-1)} = \frac{p^2(1-p^2)}{d-1}\mathcal{F}_{\mu\mu}^*. \tag{S48}$$

The second order derivative can be expressed by

$$(H_\mathcal{L})_{\mu\nu} = \frac{\partial^2\mathcal{L}}{\partial\theta_\mu\partial\theta_\nu} = \partial_\mu\partial_\nu\mathcal{L} = -\langle\phi|D_{\mu\nu}|\phi\rangle$$
$$= -p^2\langle\psi^*|D_{\mu\nu}|\psi^*\rangle - q^2\langle\psi^\perp|D_{\mu\nu}|\psi^\perp\rangle - 2pq\,\mathrm{Re}\left(\langle\psi^\perp|D_{\mu\nu}|\psi^*\rangle\right), \tag{S49}$$

where $D_{\mu\nu} = \partial_\mu\partial_\nu\varrho$ is a traceless Hermitian operator since $\mathrm{tr}\,D_{\mu\nu} = \partial_\mu\partial_\nu(\mathrm{tr}\,\varrho) = 0$. Please do not confuse $D_{\mu\nu}$ with $D_\mu$ above. At $\boldsymbol{\theta} = \boldsymbol{\theta}^*$, the $D_{\mu\nu}$ is reduced to $D_{\mu\nu}^*$ which satisfies the following properties

$$D_{\mu\nu}^* = \partial_\mu\partial_\nu(|\psi\rangle\langle\psi|)^* = 2\,\mathrm{Re}\left[|\partial_\mu\partial_\nu\psi^*\rangle\langle\psi^*| + |\partial_\mu\psi^*\rangle\langle\partial_\nu\psi^*|\right], \tag{S50}$$

$$\langle\psi^*|D_{\mu\nu}^*|\psi^*\rangle = 2\,\mathrm{Re}\left[\langle\psi^*|\partial_\mu\partial_\nu\psi^*\rangle + \langle\psi^*|\partial_\mu\psi^*\rangle\langle\partial_\nu\psi^*|\psi^*\rangle\right] = -\mathcal{F}_{\mu\nu}, \tag{S51}$$

$$\langle\psi^\perp|D_{\mu\nu}^*|\psi^\perp\rangle = 2\,\mathrm{Re}\left[\langle\psi^\perp|\partial_\nu\psi^*\rangle\langle\partial_\mu\psi^*|\psi^\perp\rangle\right]. \tag{S52}$$

Here the notation $2\,\mathrm{Re}[\cdot]$ of square matrix $A$ actually means the sum of the matrix and its Hermitian conjugate, i.e., $2\,\mathrm{Re}[A] = A + A^\dagger$. From Eq. (S50) we know that the rank of $D_{\mu\nu}$ is at most 4. Substituting the expectation in Eq. (S42), the variance of the second order derivative at $\boldsymbol{\theta} = \boldsymbol{\theta}^*$ becomes

$$\mathrm{Var}_\mathbb{T}[\partial_\mu\partial_\nu\mathcal{L}^*] = \mathbb{E}_\mathbb{T}\left[(\partial_\mu\partial_\nu\mathcal{L}^* - \mathbb{E}_\mathbb{T}[\partial_\mu\partial_\nu\mathcal{L}^*])^2\right]$$
$$= \left(\frac{q^2}{d-1}\mathcal{F}_{\mu\nu}^*\right)^2 + q^4\mathbb{E}_\mathbb{T}\left[\langle\psi^\perp|D_{\mu\nu}^*|\psi^\perp\rangle^2\right] \tag{S53}$$
$$- \frac{2q^4}{d-1}\mathcal{F}_{\mu\nu}^*\mathbb{E}_\mathbb{T}\left[\langle\psi^\perp|D_{\mu\nu}^*|\psi^\perp\rangle\right] + 4p^2q^2\mathbb{E}_\mathbb{T}\left[(\mathrm{Re}\langle\psi^\perp|D_{\mu\nu}^*|\psi^*\rangle)^2\right].$$

where the inhomogeneous cross terms vanish after taking the expectation according to Lemma S2 and have been omitted. Using Corollaries S4 and S6, the expectations in Eq. (S53) can be calculated as

$$\mathbb{E}_\mathbb{T}\left[\langle\psi^\perp|D_{\mu\nu}^*|\psi^\perp\rangle^2\right] = \frac{\mathrm{tr}((D_{\mu\nu}^*)^2) - 2\left(\langle\psi^*|(D_{\mu\nu}^*)^2|\psi^*\rangle - \langle\psi^*|D_{\mu\nu}^*|\psi^*\rangle^2\right)}{d(d-1)},$$

$$\mathbb{E}_\mathbb{T}\left[\langle\psi^\perp|D_{\mu\nu}^*|\psi^\perp\rangle\right] = -\frac{\langle\psi^*|D_{\mu\nu}^*|\psi^*\rangle}{d-1} = \frac{\mathcal{F}_{\mu\nu}^*}{d-1}, \tag{S54}$$

$$\mathbb{E}_\mathbb{T}\left[(\mathrm{Re}\langle\psi^\perp|D_{\mu\nu}^*|\psi^*\rangle)^2\right] = \frac{1}{2}\mathbb{E}_\mathbb{T}\left[\langle\psi^\perp|D_{\mu\nu}^*|\psi^*\rangle\langle\psi^*|D_{\mu\nu}^*|\psi^\perp\rangle\right]$$
$$= \frac{\langle\psi^*|(D_{\mu\nu}^*)^2|\psi^*\rangle - \langle\psi^*|D_{\mu\nu}^*|\psi^*\rangle^2}{2(d-1)}.$$

Thus the variance of the second order derivative at $\boldsymbol{\theta} = \boldsymbol{\theta}^*$ can be written as

$$\mathrm{Var}_\mathbb{T}[\partial_\mu\partial_\nu\mathcal{L}^*] = q^4\frac{\|D_{\mu\nu}^*\|_2^2 - 2\left(\langle\psi^*|(D_{\mu\nu}^*)^2|\psi^*\rangle - \langle\psi^*|D_{\mu\nu}^*|\psi^*\rangle^2\right)}{d(d-1)}$$
$$+ 2p^2q^2\frac{\langle\psi^*|(D_{\mu\nu}^*)^2|\psi^*\rangle - \langle\psi^*|D_{\mu\nu}^*|\psi^*\rangle^2}{d-1} - \left(\frac{q^2}{d-1}\mathcal{F}_{\mu\nu}^*\right)^2. \tag{S55}$$

Note that the factor

$$\langle\psi^*|(D_{\mu\nu}^*)^2|\psi^*\rangle - \langle\psi^*|D_{\mu\nu}^*|\psi^*\rangle^2 = \langle\psi^*|D_{\mu\nu}^*(I - |\psi^*\rangle\langle\psi^*|)D_{\mu\nu}^*|\psi^*\rangle, \tag{S56}$$

is non-negative because the operator $(I - |\psi^*\rangle\langle\psi^*|)$ is positive semidefinite. Hence the variance can be upper bounded by

$$
\begin{aligned}
\mathrm{Var}_{\mathbb{T}}[\partial_\mu\partial_\nu\mathcal{L}^*] &\leq \frac{q^4}{d(d-1)}\|D_{\mu\nu}^*\|_2^2 + \frac{2p^2q^2}{d-1}\langle\psi^*|(D_{\mu\nu}^*)^2|\psi^*\rangle \\
&\leq \frac{q^4}{d(d-1)}\|D_{\mu\nu}^*\|_2^2 + \frac{2p^2q^2}{d-1}\|(D_{\mu\nu}^*)^2\|_\infty - \left(\frac{q^2}{d-1}\mathcal{F}_{\mu\nu}^*\right)^2 \\
&\leq \frac{2q^2}{d-1}\left(p^2 + \frac{2q^2}{d}\right)\|D_{\mu\nu}^*\|_\infty^2,
\end{aligned}
\tag{S57}
$$

where we have used the properties

$$
\|D_{\mu\nu}^*\|_2 \leq \sqrt{\mathrm{rank}(D_{\mu\nu}^*)}\|D_{\mu\nu}^*\|_\infty \leq 2\|D_{\mu\nu}^*\|_\infty, \quad \|(D_{\mu\nu}^*)^2\|_\infty = \|D_{\mu\nu}^*\|_\infty^2.
\tag{S58}
$$

Utilizing the quantum gates in the QNN, the operator $D_{\mu\nu}$ can be written as

$$
D_{\mu\nu} = V_{\nu+1\to M}[V_{\mu+1\to\nu}[V_{1\to\mu}|0\rangle\langle0|V_{1\to\mu}^\dagger, i\Omega_\mu]V_{\mu+1\to\nu}^\dagger, i\Omega_\nu]V_{\nu+1\to M}^\dagger,
\tag{S59}
$$

where we assume $\mu \leq \nu$ without loss of generality. Thus $\|D_{\mu\nu}\|_\infty$ can be upper bounded by

$$
\|D_{\mu\nu}\|_\infty \leq 4\|\Omega_\mu\Omega_\nu\|_\infty \leq 4\|\Omega_\mu\|_\infty\|\Omega_\nu\|_\infty.
\tag{S60}
$$

Finally, the variance of the second order derivative at $\boldsymbol{\theta} = \boldsymbol{\theta}^*$ can be bounded as

$$
\mathrm{Var}_{\mathbb{T}}[\partial_\mu\partial_\nu\mathcal{L}^*] \leq f_2(p,d)\|\Omega_\mu\|_\infty^2\|\Omega_\nu\|_\infty^2.
\tag{S61}
$$

The factor $f_2(p,d)$ reads

$$
f_2(p,d) = \frac{32(1-p^2)}{d-1}\left[p^2 + \frac{2(1-p^2)}{d}\right],
\tag{S62}
$$

which vanishes at least of order $1/d$. ∎

Note that when $p \in \{0,1\}$, $f_1$, $f_2$ and hence the variances of the first and second order derivatives become exactly zero, indicating $\mathcal{L}^*$ takes the optimum in all cases. This is nothing but the fact that the range of the loss function is $[0,1]$, which reflects that the bound of $\mathrm{Var}_{\mathbb{T}}[\partial_\mu\partial_\nu\mathcal{L}^*]$ is tight in $p$.

We remark that the vanishing gradient here is both conceptually and technically distinct from barren plateaus [35]. Firstly, here we focus on a fixed parameter point $\boldsymbol{\theta}^*$ instead of a randomly chosen point on the training landscape. Other points apart from $\boldsymbol{\theta}^*$ is allowed to have a non-vanishing gradient expectation, which leads to prominent local minima instead of plateaus. Moreover, the ensemble $\mathbb{T}$ used here originates from the unknown target state instead of the random initialization. The latter typically demands a polynomially deep circuit to form a 2-design. Technically, a constant overlap $p$ is assumed to construct the ensemble $\mathbb{T}$ instead of completely random over the entire Hilbert space. Thus our results apply to adaptive methods, while barren plateaus from the random initialization are not.

**Theorem 2** *If the fidelity loss function satisfies $\mathcal{L}(\boldsymbol{\theta}^*) < 1 - 1/d$, the probability that $\boldsymbol{\theta}^*$ is not a local minimum of $\mathcal{L}$ up to a fixed precision $\epsilon = (\epsilon_1, \epsilon_2)$ with respect to the target state ensemble $\mathbb{T}$ is upper bounded by*

$$
\mathrm{Pr}_{\mathbb{T}}\left[\neg\,\mathrm{LocalMin}(\boldsymbol{\theta}^*, \epsilon)\right] \leq \frac{2f_1(p,d)\|\boldsymbol{\omega}\|_2^2}{\epsilon_1^2} + \frac{f_2(p,d)\|\boldsymbol{\omega}\|_2^4}{\left(\frac{dp^2-1}{d-1}e^* + \epsilon_2\right)^2},
\tag{S63}
$$

*where $e^*$ denotes the minimal eigenvalue of the QFI matrix at $\boldsymbol{\theta} = \boldsymbol{\theta}^*$. $f_1$ and $f_2$ are defined in Lemma 1 which vanish at least of order $1/d$.*

**Proof** By definition in Eq. (5) in the main text, the probability $\mathrm{Pr}_{\mathbb{T}}\left[\neg\,\mathrm{LocalMin}(\boldsymbol{\theta}^*, \epsilon)\right]$ can be upper bounded by the sum of two terms: the probability that one of the gradient component is larger than

$\epsilon_1$, and the probability that the Hessian matrix is not positive definite up to the error $\epsilon_2$, i.e.,

$$\Pr_{\mathbb{T}}\left[\neg\operatorname{LocalMin}(\boldsymbol{\theta}^*, \epsilon)\right] = \Pr_{\mathbb{T}}\left[\bigcup_{\mu=1}^{M}\{|\partial_\mu\mathcal{L}^*| > \epsilon_1\} \cup \{H_\mathcal{L}^* \not\succ -\epsilon_2 I\}\right]$$
$$\leq \Pr_{\mathbb{T}}\left[\bigcup_{\mu=1}^{M}\{|\partial_\mu\mathcal{L}^*| > \epsilon_1\}\right] + \Pr_{\mathbb{T}}\left[H_\mathcal{L}^* \not\succ -\epsilon_2 I\right]. \tag{S64}$$

The first term can be easily upper bounded by combining Lemma 1 and Chebyshev's inequality, i.e.,

$$\Pr_{\mathbb{T}}\left[\bigcup_{\mu=1}^{M}\{|\partial_\mu\mathcal{L}^*| > \epsilon_1\}\right] \leq \sum_{\mu=1}^{M}\Pr_{\mathbb{T}}\left[|\partial_\mu\mathcal{L}^*| > \epsilon_1\right] \leq \sum_{\mu=1}^{M}\frac{\operatorname{Var}_{\mathbb{T}}[\partial_\mu\mathcal{L}^*]}{\epsilon_1^2} = \frac{f_1(p,d)}{\epsilon_1^2}\operatorname{tr}\mathcal{F}^*, \tag{S65}$$

where the diagonal element of the QFI matrix is upper bounded as $\mathcal{F}_{\mu\mu} \leq 2\|\Omega_\mu\|_\infty^2$ by definition and thus $\operatorname{tr}\mathcal{F}^* \leq 2\|\boldsymbol{\omega}\|_2^2$. Here the generator norm vector $\boldsymbol{\omega}$ is defined as

$$\boldsymbol{\omega} = (\|\Omega_1\|_\infty, \|\Omega_2\|_\infty, \ldots, \|\Omega_M\|_\infty), \tag{S66}$$

so that the squared vector 2-norm of $\boldsymbol{\omega}$ equals to $\|\boldsymbol{\omega}\|_2^2 = \sum_{\mu=1}^{M}\|\Omega_\mu\|_\infty^2$. Thus we obtain the upper bound of the first term, i.e.,

$$\Pr_{\mathbb{T}}\left[\bigcup_{\mu=1}^{M}\{|\partial_\mu\mathcal{L}^*| > \epsilon_1\}\right] \leq \frac{2f_1(p,d)\|\boldsymbol{\omega}\|_2^2}{\epsilon_1^2}, \tag{S67}$$

It takes extra efforts to bound the second term. After assuming $p^2 > 1/d$ to ensure that $\mathbb{E}_{\mathbb{T}}[H_\mathcal{L}^*]$ is positive semidefinite, a sufficient condition of the positive definiteness can be obtained by perturbing $\mathbb{E}_{\mathbb{T}}[H_\mathcal{L}^*]$ using Lemma S8, i.e.,

$$\|H_\mathcal{L}^* - \mathbb{E}_{\mathbb{T}}[H_\mathcal{L}^*]\|_\infty < \|\mathbb{E}_{\mathbb{T}}[H_\mathcal{L}^* + \epsilon_2 I]^{-1}\|_\infty^{-1} \quad \Rightarrow \quad H_\mathcal{L}^* + \epsilon_2 I \succ 0, \tag{S68}$$

Note that $\|\mathbb{E}_{\mathbb{T}}[H_\mathcal{L}^* + \epsilon_2 I]^{-1}\|_\infty^{-1} = \frac{dp^2-1}{d-1}e^* + \epsilon_2$, where $e^*$ denotes the minimal eigenvalue of the QFI $\mathcal{F}^*$. A necessary condition for $H_\mathcal{L}^* + \epsilon_2 I \not\succ 0$ is hence obtained by the contrapositive, i.e.,

$$H_\mathcal{L}^* \not\succ -\epsilon_2 I \quad \Rightarrow \quad \|H_\mathcal{L}^* - \mathbb{E}_{\mathbb{T}}[H_\mathcal{L}^*]\|_\infty \geq \frac{dp^2-1}{d-1}e^* + \epsilon_2. \tag{S69}$$

Thus the probability that $H_\mathcal{L}^*$ is not positive definite can be upper bounded by

$$\Pr_{\mathbb{T}}\left[H_\mathcal{L}^* \not\succ -\epsilon_2 I\right] \leq \Pr_{\mathbb{T}}\left[\|H_\mathcal{L}^* - \mathbb{E}_{\mathbb{T}}[H_\mathcal{L}^*]\|_\infty \geq \frac{dp^2-1}{d-1}e^* + \epsilon_2\right]. \tag{S70}$$

The generalized Chebyshev's inequality in Lemma S11 regarding $H_\mathcal{L}^*$ and the Schatten-$\infty$ norm gives

$$\Pr_{\mathbb{T}}\left[\|H_\mathcal{L}^* - \mathbb{E}_{\mathbb{T}}[H_\mathcal{L}^*]\|_\infty \geq \varepsilon\right] \leq \frac{\sigma_\infty^2}{\varepsilon^2}, \tag{S71}$$

where the "norm variance" is defined as $\sigma_\infty^2 = \mathbb{E}_{\mathbb{T}}[\|H_\mathcal{L}^* - \mathbb{E}_{\mathbb{T}}[H_\mathcal{L}^*]\|_\infty^2]$. By taking $\varepsilon = \frac{dp^2-1}{d-1}e^* + \epsilon_2$, we obtain

$$\Pr_{\mathbb{T}}\left[H_\mathcal{L}^* \not\succeq -\epsilon_2 I\right] \leq \frac{\sigma_\infty^2}{\left(\frac{dp^2-1}{d-1}e^* + \epsilon_2\right)^2}. \tag{S72}$$

Utilizing Lemma 1, $\sigma_\infty^2$ can be further bounded by

$$\sigma_\infty^2 \leq \sigma_2^2 = \mathbb{E}_{\mathbb{T}}[\|H_\mathcal{L}^* - \mathbb{E}_{\mathbb{T}}[H_\mathcal{L}^*]\|_2^2] = \sum_{\mu\nu}\mathbb{E}_{\mathbb{T}}\left[((H_\mathcal{L}^*)_{\mu\nu} - (\mathbb{E}_{\mathbb{T}}[H_\mathcal{L}^*])_{\mu\nu})^2\right]$$
$$= \sum_{\mu,\nu=1}^{M}\operatorname{Var}_{\mathbb{T}}\left[\partial_\mu\partial_\nu\mathcal{L}^*\right] \leq f_2(p,d)\sum_{\mu,\nu=1}^{M}\|\Omega_\mu\|_\infty^2\|\Omega_\nu\|_\infty^2 \tag{S73}$$
$$= f_2(p,d)\left(\sum_{\mu=1}^{M}\|\Omega_\mu\|_\infty^2\right)^2 = f_2(p,d)\|\boldsymbol{\omega}\|_2^4.$$

Combining Eqs. (S72) and (S73), we obtain the upper bound of the second term, i.e.,

$$\Pr_{\mathbb{T}}\left[H_{\mathcal{L}}^* \not\succeq -\epsilon_2 I\right] \leq \frac{f_2(p,d)\|\boldsymbol{\omega}\|_2^4}{\left(\frac{dp^2-1}{d-1}e^* + \epsilon_2\right)^2}. \tag{S74}$$

Substituting the bounds for the first and second terms into Eq. (S64), one finally arrives at the desired upper bound for the probability that $\boldsymbol{\theta}^*$ is not a local minimum up to a fixed precision $\epsilon = (\epsilon_1, \epsilon_2)$. $\blacksquare$

Note that the conclusion can be generalized to the scenario of mixed states or noisy states easily using the mathematical tools we developed in Appendix A.1. For example, suppose that the output state of the QNN is $\rho(\boldsymbol{\theta})$ and the target state is $|\phi\rangle$. The loss function can be defined as the fidelity distance $\mathcal{L}(\boldsymbol{\theta}) = 1 - \langle\phi|\rho(\boldsymbol{\theta})|\phi\rangle$. Utilizing Lemmas S3 and S7, similar results can be carried out by calculating the subspace Haar integration.

**Proposition 3** *The expectation and variance of the fidelity loss function $\mathcal{L}$ with respect to the target state ensemble $\mathbb{T}$ can be exactly calculated as*

$$\mathbb{E}_{\mathbb{T}}\left[\mathcal{L}(\boldsymbol{\theta})\right] = 1 - p^2 + \frac{dp^2-1}{d-1}g(\boldsymbol{\theta}),$$

$$\mathrm{Var}_{\mathbb{T}}\left[\mathcal{L}(\boldsymbol{\theta})\right] = \frac{1-p^2}{d-1}g(\boldsymbol{\theta})\left[4p^2 - \left(2p^2 - \frac{(d-2)(1-p^2)}{d(d-1)}\right)g(\boldsymbol{\theta})\right], \tag{S75}$$

*where $g(\boldsymbol{\theta}) = 1 - |\langle\psi^*|\psi(\boldsymbol{\theta})\rangle|^2$.*

**Proof** The expression of the expectation $\mathbb{E}_{\mathbb{T}}[\mathcal{L}]$ has already been calculated in Eq. (S41). Considering Lemma S2, the variance of the loss function is

$$\mathrm{Var}_{\mathbb{T}}[\mathcal{L}] = \mathbb{E}_{\mathbb{T}}\left[\left(\mathcal{L} - \mathbb{E}_{\mathbb{T}}[\mathcal{L}]\right)^2\right]$$

$$= \mathbb{E}_{\mathbb{T}}\left[\left(\frac{1-p^2}{d-1}(\langle\psi^*|\varrho|\psi^*\rangle - 1) + q^2\langle\psi^\perp|\varrho|\psi^\perp\rangle + 2pq\,\mathrm{Re}\left(\langle\psi^\perp|\varrho|\psi^*\rangle\right)\right)^2\right] \tag{S76}$$

$$= \frac{q^4}{(d-1)^2}(\langle\psi^*|\varrho|\psi^*\rangle - 1)^2 + \frac{2q^4}{d-1}(\langle\psi^*|\varrho|\psi^*\rangle - 1)\,\mathbb{E}_{\mathbb{T}}\left[\langle\psi^\perp|\varrho|\psi^\perp\rangle\right]$$

$$+ q^4\,\mathbb{E}_{\mathbb{T}}\left[\langle\psi^\perp|\varrho|\psi^\perp\rangle^2\right] + 4p^2q^2\,\mathbb{E}_{\mathbb{T}}\left[\mathrm{Re}\left(\langle\psi^\perp|\varrho|\psi^*\rangle\right)^2\right],$$

where $q = \sqrt{1-p^2}$ and $\varrho(\boldsymbol{\theta}) = |\psi(\boldsymbol{\theta})\rangle\langle\psi(\boldsymbol{\theta})|$. According to Corollaries S4 and S6, the terms above can be calculated as

$$\mathbb{E}_{\mathbb{T}}\left[\langle\psi^\perp|\varrho|\psi^\perp\rangle\right] = \frac{1 - \langle\psi^*|\varrho|\psi^*\rangle}{d-1},$$

$$\mathbb{E}_{\mathbb{T}}\left[\langle\psi^\perp|\varrho|\psi^\perp\rangle^2\right] = \frac{\left(\mathrm{tr}(\varrho^2) - 2\langle\psi^*|\varrho^2|\psi^*\rangle + \langle\psi^*|\varrho|\psi^*\rangle^2\right) + (1 - \langle\psi^*|\varrho|\psi^*\rangle)^2}{d(d-1)}$$

$$= \frac{2(1 - \langle\psi^*|\varrho|\psi^*\rangle)^2}{d(d-1)}, \tag{S77}$$

$$\mathbb{E}_{\mathbb{T}}\left[\mathrm{Re}\left(\langle\psi^\perp|\varrho|\psi^*\rangle\right)^2\right] = \frac{1}{2}\mathbb{E}_{\mathbb{T}}\left[\langle\psi^\perp|\varrho|\psi^*\rangle\langle\psi^*|\varrho|\psi^\perp\rangle\right] = \frac{1 - \langle\psi^*|\varrho|\psi^*\rangle^2}{2(d-1)}.$$

Thus the variance of the loss function becomes

$$\mathrm{Var}_{\mathbb{T}}[\mathcal{L}] = -\frac{q^4(1 - \langle\psi^*|\varrho|\psi^*\rangle)^2}{(d-1)^2} + \frac{2q^4(1 - \langle\psi^*|\varrho|\psi^*\rangle)^2}{d(d-1)} + \frac{2p^2q^2(1 - \langle\psi^*|\varrho|\psi^*\rangle^2)}{d-1}$$

$$= \frac{q^2(1 - \langle\psi^*|\varrho|\psi^*\rangle)}{d-1}\left[\frac{q^2(d-2)(1 - \langle\psi^*|\varrho|\psi^*\rangle)}{d(d-1)} + 2p^2(1 + \langle\psi^*|\varrho|\psi^*\rangle)\right]. \tag{S78}$$

Substituting the relation $\langle\psi^*|\varrho|\psi^*\rangle = 1 - g(\boldsymbol{\theta})$, the desired expression is obtained. $\blacksquare$

If the quantum gate $U_\mu$ in the QNN satisfies the parameter-shift rule, the explicit form of the factor $g(\boldsymbol{\theta})$ could be known along the axis of $\theta_\mu$ passing through $\boldsymbol{\theta}^*$, which is summarized in Corollary S12. We use $\theta_{\bar{\mu}}$ to represent the other components except for $\theta_\mu$, namely $\theta_{\bar{\mu}} = \{\theta_\nu\}_{\nu\neq\mu}$.

**Corollary S12** *For QNNs satisfying the parameter-shift rule by $\Omega_\mu^2 = I$, the expectation and variance of the fidelity loss function $\mathcal{L}$ restricted by only varying the parameter $\theta_\mu$ from $\boldsymbol{\theta}^*$ with respect to the target state ensemble $\mathbb{T}$ can be exactly calculated as*

$$\mathbb{E}_{\mathbb{T}}\left[ \mathcal{L}|_{\theta_{\bar{\mu}}=\theta_{\bar{\mu}}^*} \right] = 1 - p^2 + \frac{dp^2 - 1}{d - 1}g(\theta_\mu),$$

$$\mathrm{Var}_{\mathbb{T}}\left[ \mathcal{L}|_{\theta_{\bar{\mu}}=\theta_{\bar{\mu}}^*} \right] = \frac{1 - p^2}{d - 1}g(\theta_\mu)\left[ 4p^2 - \left( 2p^2 - \frac{(d-2)(1-p^2)}{d(d-1)} \right) g(\theta_\mu) \right], \tag{S79}$$

*where $g(\theta_\mu) = \frac{1}{2}\mathcal{F}_{\mu\mu}^* \sin^2\left( \theta_\mu - \theta_\mu^* \right)$.*

**Proof** According to Proposition 3, we only need to calculate the factor $g(\boldsymbol{\theta})|_{\theta_{\bar{\mu}}=\theta_{\bar{\mu}}^*}$. We simply denote this factor as $g(\theta_\mu)$, the explicit expression of which could be calculated by just substituting the parameter-shift rule. Alternatively, the expression of $g(\theta_\mu)$ can be directly written down by considering the following facts. The parameter-shift rule ensures that $g(\theta_\mu)$ must take the form of linear combinations of 1, $\cos(2\theta_\mu)$ and $\sin(2\theta_\mu)$ since $U_\mu(\theta_\mu) = e^{-i\Omega_\mu \theta_\mu} = \cos\theta_\mu I - i\sin\theta_\mu \Omega_\mu$ and $g(\theta_\mu)$ takes the form of $U_\mu(\cdot)U_\mu^\dagger$. Furthermore, $g(\theta_\mu)$ takes its minimum at $\theta_\mu^*$ so that it is an even function relative to $\theta_\mu = \theta_\mu^*$. Combined with the fact that $g(\theta_\mu)$ also takes zero at $\theta_\mu^*$, we know $g(\theta_\mu) \propto [1 - \cos(2(\theta_\mu - \theta_\mu^*))]$. The coefficient can be determined by considering that the second order derivative of $g(\theta_\mu)$ equals to the QFI matrix element $\mathcal{F}_{\mu\mu}^*$ by definition, so that

$$g(\theta_\mu) = g(\boldsymbol{\theta})|_{\theta_{\bar{\mu}}=\theta_{\bar{\mu}}^*} = \frac{1}{4}\mathcal{F}_{\mu\mu}^*[1 - \cos(2(\theta_\mu - \theta_\mu^*))] = \frac{1}{2}\mathcal{F}_{\mu\mu}^* \sin^2(\theta_\mu - \theta_\mu^*). \tag{S80}$$

The expressions of the expectation and variance of the loss function can be obtained by directly substituting Eq. (S80) into Proposition 3. ∎

## C Generalization to the local loss function

In the main text, we focus on the fidelity loss function, also known as the "global" loss function [36], where the ensemble construction and calculation are preformed in a clear and meaningful manner. However, there is another type of loss function called "local" loss function [36], such as the energy expectation in the variational quantum eigensolver (VQE) which aims to prepare the ground state of a physical system. The local loss function takes the form of

$$\mathcal{L}(\boldsymbol{\theta}) = \langle\psi(\boldsymbol{\theta})|H|\psi(\boldsymbol{\theta})\rangle, \tag{S81}$$

where $H$ is the Hamiltonian of the physical system as a summation of Pauli strings. Eq. (S81) can formally reduce to the fidelity loss function by taking $H = I - |\phi\rangle\langle\phi|$. In this section, we generalize the results of the fidelity loss function to the local loss function and show that the conclusion keeps the same, though the ensemble construction and calculation are more complicated.

The ensemble we used in the main text decomposes the unknown target state into the learnt component $|\psi^*\rangle$ and the unknown component $|\psi^\perp\rangle$, and regards $|\psi^\perp\rangle$ as a Haar random state in the orthogonal complement of $|\psi^*\rangle$. This way of thinking seems to be more subtle in the case of the local loss function since the Hamiltonian is usually already known in the form of Pauli strings and hence it is unnatural to assume an unknown Hamiltonian. However, a known Hamiltonian does not imply a known target state, i.e., the ground state of the physical system. One needs to diagonalize the Hamiltonian to find the ground state, which requires an exponential cost in classical computers. That is to say, what one really does not know is the unitary used in the diagonalization, i.e., the relation between the learnt state $|\psi^*\rangle$ and the eigen-basis of the Hamiltonian. We represent this kind of uncertainty by a unitary $V$ from the ensemble $\mathbb{V}$, where $\mathbb{V}$ comes from the ensemble $\mathbb{U}$ mentioned in Appendix A.1 by specifying $\bar{P} = |\psi^*\rangle\langle\psi^*|$. Such an ensemble $\mathbb{V}$ induces an ensemble of loss functions via

$$\mathcal{L}(\boldsymbol{\theta}) = \langle\psi(\boldsymbol{\theta})|V^\dagger H V|\psi(\boldsymbol{\theta})\rangle, \tag{S82}$$

similar with the loss function ensemble induced by the unknown target state in the main text. $\mathbb{V}$ can be interpreted as all of the possible diagonalizing unitaries that keeps the loss value $\mathcal{L}(\boldsymbol{\theta}^*)$ constant, denoted as $\mathcal{L}^*$. In the following, similar with those for the global loss function, we calculate the expectation and variance of the derivatives of the local loss function in Lemma S13 and bound the probability of avoiding local minima in Theorem S14. Hence, the results and relative discussions in the main text could generalize to the case of local loss functions.

**Lemma S13** *The expectation and variance of the gradient $\nabla \mathcal{L}$ and Hessian matrix $H_{\mathcal{L}}$ of the local loss function $\mathcal{L}(\boldsymbol{\theta}) = \langle \psi(\boldsymbol{\theta})|H|\psi(\boldsymbol{\theta})\rangle$ at $\boldsymbol{\theta} = \boldsymbol{\theta}^*$ with respect to the ensemble $\mathbb{V}$ satisfy*

$$\mathbb{E}_{\mathbb{V}}[\nabla \mathcal{L}^*] = 0, \quad \mathrm{Var}_{\mathbb{V}}[\partial_\mu \mathcal{L}^*] = f_1(H, d)\mathcal{F}_{\mu\mu}^*,$$

$$\mathbb{E}_{\mathbb{V}}[H_{\mathcal{L}}^*] = \frac{\mathrm{tr}\, H - d\mathcal{L}^*}{d-1}\mathcal{F}^*, \quad \mathrm{Var}_{\mathbb{V}}[\partial_\mu \partial_\nu \mathcal{L}^*] \leq f_2(H, d)\|\Omega_\mu\|_\infty^2 \|\Omega_\nu\|_\infty^2, \tag{S83}$$

*where $\mathcal{F}$ denotes the QFI matrix. $f_1$ and $f_2$ are functions of the Hamiltonian $H$ and the Hilbert space dimension $d$, i.e.,*

$$f_1(H, d) = \frac{\langle H^2 \rangle_* - \langle H \rangle_*^2}{d-1}, \quad f_2(H, d) = 32\left(\frac{\langle H^2 \rangle_* - \langle H \rangle_*^2}{d-1} + \frac{2\|H\|_2^2}{d(d-2)}\right), \tag{S84}$$

*where we introduce the notation $\langle \cdot \rangle_* = \langle \psi^*| \cdot |\psi^* \rangle$.*

**Proof** Using Lemma S3, the expectation of the local loss function can be directly calculated as

$$\mathbb{E}_{\mathbb{V}}[\mathcal{L}(\boldsymbol{\theta})] = \mathcal{L}^* + \frac{\mathrm{tr}\, H - d\mathcal{L}^*}{d-1}g(\boldsymbol{\theta}). \tag{S85}$$

where $g(\boldsymbol{\theta}) = 1 - \langle \psi^*|\varrho(\boldsymbol{\theta})|\psi^* \rangle$ denotes the fidelity distance between the output states at $\boldsymbol{\theta}$ and $\boldsymbol{\theta}^*$. By definition, $g(\boldsymbol{\theta})$ takes the global minimum at $\boldsymbol{\theta} = \boldsymbol{\theta}^*$. Thus the commutation between the expectation and differentiation gives

$$\mathbb{E}_{\mathbb{V}}[\nabla \mathcal{L}^*] = \nabla\left(\mathbb{E}_{\mathbb{V}}[\mathcal{L}]\right)|_{\boldsymbol{\theta}=\boldsymbol{\theta}^*} = \frac{\mathrm{tr}\, H - d\mathcal{L}^*}{d-1}\nabla g(\boldsymbol{\theta})|_{\boldsymbol{\theta}=\boldsymbol{\theta}^*} = 0,$$

$$\mathbb{E}_{\mathbb{V}}[H_{\mathcal{L}}^*] = \frac{\mathrm{tr}\, H - d\mathcal{L}^*}{d-1}H_g(\boldsymbol{\theta})|_{\boldsymbol{\theta}=\boldsymbol{\theta}^*} = \frac{\mathrm{tr}\, H - d\mathcal{L}^*}{d-1}\mathcal{F}^*. \tag{S86}$$

By definition, $H_g(\boldsymbol{\theta})|_{\boldsymbol{\theta}=\boldsymbol{\theta}^*}$ is actually the QFI matrix $\mathcal{F}^*$ of $|\psi(\boldsymbol{\theta})\rangle$ at $\boldsymbol{\theta} = \boldsymbol{\theta}^*$ (see Appendix A.4), which is always positive semidefinite. To estimate the variance, we need to calculate the expression of derivatives first due to the non-linearity of the variance. The first order derivative of the local loss function can be expressed by

$$\partial_\mu \mathcal{L} = \mathrm{tr}[V^\dagger H V D_\mu] = 2\,\mathrm{Re}\langle \psi|V^\dagger H V|\partial_\mu \psi\rangle, \tag{S87}$$

where $D_\mu = \partial_\mu \varrho$ is a traceless Hermitian operator since $\mathrm{tr}\, D_\mu = \partial_\mu(\mathrm{tr}\, \varrho) = 0$. By definition, we know that $|\psi\rangle^*$ is not changed by $V$, i.e., $V|\psi^*\rangle = |\psi^*\rangle$, which leads to the reduction $\partial_\mu \mathcal{L}^* = 2\,\mathrm{Re}(\langle \psi^*|HV|\partial_\mu \psi^*\rangle)$. Hence, the variance of the first order derivative at $\boldsymbol{\theta} = \boldsymbol{\theta}^*$ is

$$\begin{aligned}
\mathrm{Var}_{\mathbb{V}}[\partial_\mu \mathcal{L}^*] &= \mathbb{E}_{\mathbb{V}}\left[(\partial_\mu \mathcal{L}^* - \mathbb{E}_{\mathbb{V}}[\partial_\mu \mathcal{L}^*])^2\right] = \mathbb{E}_{\mathbb{V}}\left[(\partial_\mu \mathcal{L}^*)^2\right] \\
&= \mathbb{E}_{\mathbb{V}}\left[(2\,\mathrm{Re}\langle \psi^*|HV|\partial_\mu \psi^*\rangle)^2\right] \\
&= \mathbb{E}_{\mathbb{V}}\left[\langle \psi^*|HV|\partial_\mu \psi^*\rangle^2\right] + \mathbb{E}_{\mathbb{V}}\left[\langle \partial_\mu \psi^*|V^\dagger H|\psi^*\rangle^2\right] \\
&\quad + 2\mathbb{E}_{\mathbb{V}}\left[\langle \psi^*|HV|\partial_\mu \psi^*\rangle\langle \partial_\mu \psi^*|V^\dagger H|\psi^*\rangle\right].
\end{aligned} \tag{S88}$$

Utilizing Lemmas S2 and S3, we obtain

$$\begin{aligned}
\mathbb{E}_{\mathbb{V}}\left[\langle \psi^*|HV|\partial_\mu \psi^*\rangle^2\right] &= \langle H \rangle_*^2 \langle \psi^*|\partial_\mu \psi^*\rangle^2, \\
\mathbb{E}_{\mathbb{V}}\left[\langle \partial_\mu \psi^*|V^\dagger H|\psi^*\rangle^2\right] &= \langle H \rangle_*^2 \langle \partial_\mu \psi^*|\psi^*\rangle^2, \\
\mathbb{E}_{\mathbb{V}}\left[\langle \psi^*|HV|\partial_\mu \psi^*\rangle\langle \partial_\mu \psi^*|V^\dagger H|\psi^*\rangle\right] & \\
&\hspace{-3cm}= \frac{\langle H^2 \rangle_* - \langle H \rangle_*^2}{2(d-1)}\mathcal{F}_{\mu\mu}^* + \langle H \rangle_*^2 \langle \psi^*|\partial_\mu \psi^*\rangle\langle \partial_\mu \psi^*|\psi^*\rangle,
\end{aligned} \tag{S89}$$

where we introduce the notation $\langle \cdot \rangle_* = \langle \psi^*| \cdot |\psi^* \rangle$ and hence $\mathcal{L}^* = \langle H \rangle_*$. The $1/2$ factor in the third line arises from the definition of the QFI matrix. Note that there are three terms above canceling each other due to the fact

$$\begin{aligned}
2\,\mathrm{Re}\left[\langle \partial_\mu \psi|\psi\rangle\right] &= \langle \partial_\mu \psi|\psi\rangle + \langle \psi|\partial_\mu \psi\rangle = \partial_\mu(\langle \psi|\psi\rangle) = 0, \\
\langle \psi|\partial_\mu \psi\rangle^2 + \langle \partial_\mu \psi|\psi\rangle^2 + 2\langle \psi|\partial_\mu \psi\rangle\langle \partial_\mu \psi|\psi\rangle &= (2\,\mathrm{Re}\left[\langle \partial_\mu \psi|\psi\rangle\right])^2 = 0.
\end{aligned} \tag{S90}$$

Therefore, the variance of the first order derivative at $\boldsymbol{\theta} = \boldsymbol{\theta}^*$ equals to

$$\text{Var}_{\mathbb{V}}\left[\partial_\mu \mathcal{L}^*\right] = \mathbb{E}_{\mathbb{V}}\left[(2\,\text{Re}\langle\psi^*|HV|\partial_\mu\psi^*\rangle)^2\right] = \frac{\langle H^2\rangle_* - \langle H\rangle_*^2}{d-1}\mathcal{F}^*_{\mu\mu}. \tag{S91}$$

The second-order derivative can be expressed by

$$\partial_\mu\partial_\nu\mathcal{L} = (H_\mathcal{L})_{\mu\nu} = \text{tr}\left[V^\dagger HVD_{\mu\nu}\right], \tag{S92}$$

where $D_{\mu\nu} = \partial_\mu\partial_\nu\varrho$ is a traceless Hermitian operator since $\text{tr}\,D_{\mu\nu} = \partial_\mu\partial_\nu(\text{tr}\,\varrho) = 0$. By direct expansion, the variance of the second order derivative at $\boldsymbol{\theta} = \boldsymbol{\theta}^*$ can be expressed as

$$\text{Var}_{\mathbb{V}}[\partial_\mu\partial_\nu\mathcal{L}^*] = \mathbb{E}_{\mathbb{V}}\left[(\partial_\mu\partial_\nu\mathcal{L}^*)^2\right] - (\mathbb{E}_{\mathbb{V}}\left[\partial_\mu\partial_\nu\mathcal{L}^*\right])^2, \tag{S93}$$

where the second term is already obtained in Eq. (S86). Lemma S7 directly implies

$$\begin{aligned}
\mathbb{E}_{\mathbb{V}}\left[(\partial_\mu\partial_\nu\mathcal{L})^2\right] &= \mathbb{E}_{\mathbb{V}}\left[\text{tr}(V^\dagger HVD_{\mu\nu})\,\text{tr}(V^\dagger HVD_{\mu\nu})\right] \\
&= \langle H\rangle_*^2\langle D_{\mu\nu}\rangle_*^2 + \frac{2\langle H\rangle_*\langle D_{\mu\nu}\rangle_*}{d-1}(\text{tr}\,H - \langle H\rangle_*)(\text{tr}\,D_{\mu\nu} - \langle D_{\mu\nu}\rangle_*) \\
&\quad + \frac{2}{d-1}(\langle H^2\rangle_* - \langle H\rangle_*^2)(\langle D_{\mu\nu}^2\rangle_* - \langle D_{\mu\nu}\rangle_*^2) \\
&\quad + \frac{1}{d(d-2)}(\text{tr}\,H - \langle H\rangle_*)^2(\text{tr}\,D_{\mu\nu} - \langle D_{\mu\nu}\rangle_*)^2 \\
&\quad + \frac{1}{d(d-2)}(\text{tr}(H^2) - 2\langle H^2\rangle_* + \langle H\rangle_*^2)(\text{tr}(D_{\mu\nu}^2) - 2\langle D_{\mu\nu}^2\rangle_* + \langle D_{\mu\nu}\rangle_*^2) \\
&\quad - \frac{1}{d(d-1)(d-2)}\left[(\text{tr}(H^2) - 2\langle H^2\rangle_* + \langle H\rangle_*^2)(\text{tr}\,D_{\mu\nu} - \langle D_{\mu\nu}\rangle_*)^2\right] \\
&\quad - \frac{1}{d(d-1)(d-2)}\left[(\text{tr}\,H - \langle H\rangle_*)^2(\text{tr}(D_{\mu\nu}^2) - 2\langle D_{\mu\nu}^2\rangle_* + \langle D_{\mu\nu}\rangle_*^2)\right].
\end{aligned} \tag{S94}$$

According to Eq. (S86), $\langle D_{\mu\nu}^*\rangle_* = -\mathcal{F}^*_{\mu\nu}$ in Eq. (S51) and $\mathcal{L}^* = \langle H\rangle_*$, we have

$$(\mathbb{E}_{\mathbb{V}}[\partial_\mu\partial_\nu\mathcal{L}^*])^2 = \left(\frac{\text{tr}\,H - d\mathcal{L}^*}{d-1}\mathcal{F}^*_{\mu\nu}\right)^2 = \left(\frac{\text{tr}\,H - \langle H\rangle_*}{d-1} - \langle H\rangle_*\right)^2\langle D_{\mu\nu}^*\rangle_*^2. \tag{S95}$$

Combining Eqs. (S94) and (S95) together with the condition $\text{tr}\,D_{\mu\nu} = 0$, we obtain

$$\begin{aligned}
\text{Var}_{\mathbb{V}}[\partial_\mu\partial_\nu\mathcal{L}^*] &= \frac{2}{d-1}(\langle H^2\rangle_* - \langle H\rangle_*^2)(\langle D_{\mu\nu}^{2*}\rangle_* - \langle D_{\mu\nu}^*\rangle_*^2) \\
&\quad + \frac{1}{d(d-2)}(\text{tr}(H^2) - 2\langle H^2\rangle_* + \langle H\rangle_*^2)(\text{tr}(D_{\mu\nu}^{2*}) - 2\langle D_{\mu\nu}^{2*}\rangle_* + \langle D_{\mu\nu}^*\rangle_*^2) \\
&\quad - \frac{1}{d(d-1)(d-2)}\left[(\text{tr}(H^2) - 2\langle H^2\rangle_* + \langle H\rangle_*^2)\langle D_{\mu\nu}^*\rangle_*^2\right] \\
&\quad - \frac{1}{d(d-1)(d-2)}\left[(\text{tr}\,H - \langle H\rangle_*)^2(\text{tr}(D_{\mu\nu}^{2*}) - 2\langle D_{\mu\nu}^{2*}\rangle_* + \frac{d-2}{d-1}\langle D_{\mu\nu}\rangle_*^2))\right].
\end{aligned} \tag{S96}$$

Note that we always have $d \geq 2$ in qubit systems. If $\text{rank}\,H \geq 2$, then it holds that

$$\text{tr}(H^2) - 2\langle H^2\rangle_* + \langle H\rangle_*^2 \geq \|H\|_2^2 - 2\|H\|_\infty^2 \geq (\text{rank}\,H)\|H\|_\infty^2 - 2\|H\|_\infty^2 \geq 0. \tag{S97}$$

Otherwise if $\text{rank}\,H = 1$ (the case of $\text{rank}\,H = 0$ is trivial), then we assume $H = \lambda|\phi\rangle\langle\phi|$ and it holds that

$$\text{tr}(H^2) - 2\langle H^2\rangle_* + \langle H\rangle_*^2 = \lambda^2 - 2\lambda^2|\langle\psi^*|\phi\rangle|^2 + \lambda^2|\langle\psi^*|\phi\rangle|^4 = \lambda^2\left(1 - |\langle\psi^*|\phi\rangle|^2\right)^2 \geq 0. \tag{S98}$$

Hence we conclude that it always holds that

$$\text{tr}(H^2) - 2\langle H^2\rangle_* + \langle H\rangle_*^2 \geq 0. \tag{S99}$$

Similarly, because $\text{tr}\,D_{\mu\nu} = 0$, we know $\text{rank}\,D_{\mu\nu} \geq 2$ and thus

$$\text{tr}(D_{\mu\nu}^2) - 2\langle D_{\mu\nu}^2\rangle_* \geq (\text{rank}\,D_{\mu\nu})\|D_{\mu\nu}\|_\infty^2 - 2\|D_{\mu\nu}\|_\infty^2 \geq 0. \tag{S100}$$

Therefore, we can upper bound the variance by just discarding the last two terms in Eq. (S96)

$$
\begin{aligned}
\mathrm{Var}_{\mathbb{V}}[\partial_\mu \partial_\nu \mathcal{L}^*] &\leq \frac{2}{d-1}(\langle H^2 \rangle_* - \langle H \rangle_*^2)(\langle D_{\mu\nu}^{2*} \rangle_* - \langle D_{\mu\nu}^* \rangle_*^2) \\
&+ \frac{1}{d(d-2)}(\mathrm{tr}(H^2) - 2\langle H^2 \rangle_* + \langle H \rangle_*^2)(\mathrm{tr}(D_{\mu\nu}^{2*}) - 2\langle D_{\mu\nu}^{2*} \rangle_* + \langle D_{\mu\nu}^* \rangle_*^2).
\end{aligned}
\tag{S101}
$$

On the other hand, we have

$$
\mathrm{tr}(H^2) - 2\langle H^2 \rangle_* + \langle H \rangle_*^2 = \mathrm{tr}(H^2) - \langle H^2 \rangle_* - (\langle H^2 \rangle_* - \langle H \rangle_*^2) \leq \mathrm{tr}(H^2),
\tag{S102}
$$

since $\langle H^2 \rangle_* - \langle H \rangle_*^2 = \langle H(I - |\psi^*\rangle\langle\psi^*|)H \rangle_* \geq 0$. A similar inequality also holds for $D_{\mu\nu}$. Thus the variance can be further bounded by

$$
\mathrm{Var}_{\mathbb{V}}[\partial_\mu \partial_\nu \mathcal{L}^*] \leq \frac{2}{d-1}(\langle H^2 \rangle_* - \langle H \rangle_*^2)\|D_{\mu\nu}^*\|_\infty^2 + \frac{4\|H\|_2^2 \|D_{\mu\nu}^*\|_\infty^2}{d(d-2)},
\tag{S103}
$$

where we have used the properties in Eq. (S58). Using the inequality in Eq. (S60) associated with the gate generators, the variance of the second order derivative at $\boldsymbol{\theta} = \boldsymbol{\theta}^*$ can be ultimately upper bounded by

$$
\mathrm{Var}_{\mathbb{V}}[\partial_\mu \partial_\nu \mathcal{L}^*] \leq f_2(H, d)\|\Omega_\mu\|_\infty^2 \|\Omega_\nu\|_\infty^2,
\tag{S104}
$$

The factor $f_2(H, d)$ reads

$$
f_2(H, d) = 32 \left( \frac{\langle H^2 \rangle_* - \langle H \rangle_*^2}{d-1} + \frac{2\|H\|_2^2}{d(d-2)} \right),
\tag{S105}
$$

which vanishes at least of order $\mathcal{O}(\mathrm{poly}(N)2^{-N})$ with the qubit count $N = \log_2 d$ if $\|H\|_\infty \in \mathcal{O}(\mathrm{poly}(N))$. ∎

**Theorem S14** *If $\mathcal{L}^* < \frac{\mathrm{tr}\, H}{d}$, the probability that $\boldsymbol{\theta}^*$ is not a local minimum of the local cost function $\mathcal{L}$ up to a fixed precision $\epsilon = (\epsilon_1, \epsilon_2)$ with respect to the ensemble $\mathbb{V}$ is upper bounded by*

$$
\mathrm{Pr}_{\mathbb{V}} \left[ \neg \mathrm{LocalMin}(\boldsymbol{\theta}^*, \epsilon) \right] \leq \frac{2f_1(H, d)\|\boldsymbol{\omega}\|_2^2}{\epsilon_1^2} + \frac{f_2(H, d)\|\boldsymbol{\omega}\|_2^4}{\left( \frac{\mathrm{tr}\, H - d\mathcal{L}^*}{d-1} e^* + \epsilon_2 \right)^2},
\tag{S106}
$$

*where $e^*$ denotes the minimal eigenvalue of the QFI matrix at $\boldsymbol{\theta} = \boldsymbol{\theta}^*$. $f_1$ and $f_2$ are defined in Eq. (S84) which vanish at least of order $\mathcal{O}(\mathrm{poly}(N)2^{-N})$ with the qubit count $N = \log_2 d$ if $\|H\|_\infty \in \mathcal{O}(\mathrm{poly}(N))$.*

**Proof** Utilizing Lemma S13, the proof is exactly the same as that of Theorem 2 up to the different hessian expectation $\mathbb{E}_{\mathbb{V}}[H_{\mathcal{L}}^*]$ and coefficient functions $f_1$ and $f_2$. ∎

