# OpenReview forum: "Statistical Analysis of Quantum State Learning Process in Quantum Neural Networks"
_NeurIPS.cc/2023/Conference — NeurIPS 2023 poster_

### Official Review · Reviewer_KReL · 2023-07-05

**Soundness:** 3 good
**Presentation:** 3 good
**Contribution:** 2 fair
**Rating:** 6
**Confidence:** 4

**Summary:**

This paper describes a formalization and experimental verification of the thesis that a quantum state $\vert\phi\rangle$ is the local minimum in the process of the QNN training. The paper is well written and provides all necessary support documents for its understanding. This work is an extension and is complementary to the work on the observed plateaus in the learning landscape during the training of QNN.

The most interesting finding is that the probability that a state $\vert\phi\rangle$ being the local minimum is inversely proportional to the number of qubits and proportionally growing with the depth of the QNN (number of layers in the network). This seems to be in opposition with the original work "Barren plateaus in quantum neural network training landscapes" where the learnability decreases with the number of qubits in the state.

The formal description seems to uphold the hypothesis.

**Strengths:**

- Problem description and formalization
- The experimental verification of the observed phenomenon


**Weaknesses:**

- The main weakness is the significance of the result. While the result is interesting and proven, it is a bit expected. In particular, the result shows us that with increasing embedding i.e. the number of parameters the representational power increases while it also vanishes with increasing number of qubits. So the first conclusion is not surprising and the second conclusion seems to follow the previous works. However, more importantly, because this papers concerns an unknown state, a conclusion should be drawn if this effect can be avoided at all. Because independently of this initial setting an unknown state can occur.

**Questions:**

- What are the more formal conclusions beyond a simple better initialization or problem aware initialization? Because your paper concerns a quite serious issue a p@rediciable discussion should be provided.
- Is there a ratio between the depth $d$ and $n$ so that the occurrence of the local minima is minimized? What is the order of such ratio?

**Limitations:**

- The paper should be better discussed as for the consequences and unique conclusions. While there is a considerable amount of explanation I feel the authors failed to provide details on how to avoid the observed effect

---

> ### Author Rebuttal · Authors · 2023-08-09
>
> We thank the reviewer for their time and recognition of our work as interesting and technically sound. A detailed response to the reviewer's comments and questions is provided below in a point-by-point manner.
>
> >$\textbf{Comment 1:}$ ``The main weakness is the significance of the result. While the result is interesting and proven, it is a bit expected. In particular, the result shows us that with increasing embedding i.e. the number of parameters the representational power increases while it also vanishes with increasing number of qubits. So the first conclusion is not surprising and the second conclusion seems to follow the previous works. However, more importantly, because this paper concerns an unknown state, a conclusion should be drawn if this effect can be avoided at all. Because independently of this initial setting an unknown state can occur.''
>
> $\textbf{Re 1:}$ Thanks for the comments. In general, our no-go theorem can be regarded as a no-free-lunch (NFL) theorem in quantum state learning tasks, which implies that the probability of avoiding local minima in learning an unknown state always vanishes exponentially with the qubit count, while only grows polynomially with the circuit depth. Although this conclusion seems partially intuitive, we emphasize that the formal statement and rigorous proof is non-trivial because we prove that the theorem holds for any initial state and we develop a complete mathematical toolkit to accomplish the proof (see ``subspace integration'' in Appendix A). Moreover, our results are distinct from previous works since our results place crucial theoretical limits for adaptive training methods, which is beyond the scope of barren plateaus [1]. Finally, we have discussed that prior information can indeed help to reduce the local minimum phenomenon in Section 3.3 (line 240-247). We will make this point more clear in the revised version of our manuscript.
>
> [1] Jarrod R. McClean, Sergio Boixo, Vadim N. Smelyanskiy, Ryan Babbush, and Hartmut Neven. Barren plateaus in quantum neural network training landscapes. Nature Communications, 9(1):1–7, mar 2018.
>
> >$\textbf{Comment 2:}$ ``What are the more formal conclusions beyond a simple better initialization or problem aware initialization? Because your paper concerns a quite serious issue a prediciable discussion should be provided.''
>
> $\textbf{Re 2:}$ Thanks for this good question. Our results establish a rigorous limit for training methods without prior information, especially for those beyond the reach of barren plateaus such as simple good initial guesses and plain adaptive methods. Hence, our results suggest that a problem-inspired QNN architacture taking advantage of prior knowledge of the target state is vitally necessary, as discussed in Section 3.3. Specifically, an example of prior knowledge from quantum many-body physics is the tensor network states [1] satisfying the entanglement area law, which lives only in a polynomially large space but generally can not be solved in two and higher spatial dimensions by classical computers. Other examples include the UCCSD ansatz [2] in quantum chemistry which utilizes the fact that the Hamiltonians usually contains few-body interactions, and the QAOA ansatz [3] in combinatorial optimization which makes use of the quantum adiabatic evolution. The ways of leveraging prior information are diverse and dependent on specific problems, necessitating further research in the future. We will enhance the clarity of this aspect in the revision.
>
> [1]. Felser, Timo, Simone Notarnicola, and Simone Montangero. Efficient tensor network ansatz for high-dimensional quantum many-body problems. Physical Review Letters 126.17:170603, 2021.
>
> [2]. Jonathan Romero, Ryan Babbush, Jarrod R McClean, Cornelius Hempel, Peter J Love, and
> Alán Aspuru-Guzik. Strategies for quantum computing molecular energies using the unitary
> coupled cluster ansatz. Quantum Science and Technology, 4(1):014008, 2018.
>
> [3]. Farhi, Edward, Jeffrey Goldstone, and Sam Gutmann. A quantum approximate optimization algorithm. arXiv preprint arXiv:1411.4028 2014.
>
> >$\textbf{Comment 3:}$ ``Is there a ratio between the depth $d$ and $n$ so that the occurrence of the local minima is minimized? What is the order of such ratio?''
>
> $\textbf{Re 3:}$ Thanks for this constructive question. According to the scaling $\mathcal{O}(N^22^{-N}D^2/\epsilon^2)$ established in our manuscript, the desired critical value of the ratio $\lambda=N/D$ indeed exists if we rewritten the scaling as $\mathcal{O}(\lambda^22^{-D\lambda}D^4/\epsilon^2)$ and fix the depth $D$. Through a straightforward differentiation, one can find the critical value of the ratio $\lambda_c$ is of order $\mathcal{O}(D^{-1})$, which can also be seen from the original scaling where there is a maximum at $N_c$ of order $\mathcal{O}(1)$. This reflects the fact that the linear growth of training parameters in each layer with the qubit count can rarely improve the exponential suppression from the lack of prior information about the target state which lives in the exponentially large Hilbert space.

---

> > ### Comment · Reviewer_KReL · 2023-08-20
> >
> > Thank you for answers, I have no further questions

---

### Official Review · Reviewer_atEi · 2023-07-05

**Soundness:** 3 good
**Presentation:** 3 good
**Contribution:** 3 good
**Rating:** 7
**Confidence:** 3

**Summary:**

The authors present a no-go theorem that reveals the limitations of learning unknown quantum states using QNNs, even with high-quality initial states. They prove that the probability of avoiding local minima decreases exponentially with the number of qubits but grows polynomially with circuit depth. The curvature of local minima is determined by the quantum Fisher information and a loss-dependent constant. These findings provide insights into the role of prior information and the scalability of QNNs, impacting their learnability and effectiveness.

**Strengths:**

The work explores the trainability of quantum neural networks. The theoretical findings give the limits on the learnability of QNN in general cases which provide some insight into the development of QNN in future studies.

**Weaknesses:**



**Questions:**

1. In this work, it focus on learning pure states through QNN, and whether the statement is also true for the mixed states.
2. Whether the no-go theorem is also true for other loss functions, such as using another metric as the distance in loss?
3. In the numerical experiments, how to calculate the probability $Pr_{\mathbb{T}}[LocalMin(\theta^*,\epsilon)]$?

**Limitations:**

---

> ### Author Rebuttal · Authors · 2023-08-09
>
> We greatly appreciate the reviewer's recognition of our work as a technically solid paper with good impact. Below is out point-by-point response to the reviewer's questions.
>
> >$\textbf{Comment 1:}$ "In this work, it focus on learning pure states through QNN, and whether the statement is also true for the mixed states.''
>
> $\textbf{Re 1:}$ Thanks a lot for this highly practical question. In the main text, we indeed focused on the case of pure states for the sake of simplicity. Understanding the learning of pure quantum states is central to quantum state learning.
> However, we want to remark that the mathematical tools we developed in Appendix A can be also applied to the scenario of mixed or noisy output states, leading to similar conclusions. For example, suppose that the output state of the QNN is $\rho(\boldsymbol{\theta})$ and the target state is $|\phi\rangle$. The loss function can be defined as the fidelity distance $\mathcal{L}(\boldsymbol{\theta})=1-\langle\phi|\rho(\boldsymbol{\theta})|\phi\rangle$. Utilizing Lemmas S3 and S7, similar results can be carried out by calculating the subspace Haar integration. We shall add this generalization in the revised version of our manuscript. Nevertheless, when the output state and the target state are both mixed states, our theoretical tools can not be directly applied for the following reasons. First, the Bures fidelity of two mixed states $F(\rho,\sigma)=\operatorname{tr}\sqrt{\rho^{1/2}\sigma\rho^{1/2}}$ is hard to measure accurately on quantum devices. Second, for two mixed states, it is subtle and unclear to define the orthogonal decomposition like in Eq. (4), i.e., decompose the target state into the learned component and the unknown component, which may be left for future research.
>
> >$\textbf{Comment 2:}$ "Whether the no-go theorem is also true for other loss functions, such as using another metric as the distance in loss?''
>
> $\textbf{Re 2:}$ This is a very good question. Our theorem is indeed also true for certain other loss functions, such as the energy loss function (the expectation value of a Hermitian operator, or say the local loss function) in variational quantum eigen-solvers or combinatorial problems as shown in Appendix C. However, extending the results to other loss functions, such as alternative distance metrics, is not as straightforward and requires careful case-by-case proof utilizing the mathematical tools we developed in Appendix A, because the bound analysis for the Hessian matrix depends on the specific choice of loss functions. Nonetheless, here we can provide a general statement of the insight behind the rigorous results. The loss value below the average level actually indicates that the current parameters are relatively good. At this point, moving in a random optimization direction is likely to be less beneficial than staying in place, suggesting the presence of a local minimum. The probability of avoiding such a local minimum should be proportional to the ratio of the number of selected directions (training parameters) over the total number of all possible directions (Hilbert space dimension). Thus, the probability should in principle decay exponentially with the number of qubits, while grows polynomially with the number of training parameters.
>
> >$\textbf{Comment 3:}$ "In the numerical experiments, how to calculate the probability $\operatorname{Pr}_{\mathbb{T}} [\operatorname{LocalMin}(\boldsymbol{\theta}^* , \epsilon)]$?''
>
> $\textbf{Re 3:}$ Thanks for this careful question. In general, we numerically estimate the desired probability $\operatorname{Pr}_{\mathbb{T}} [\operatorname{LocalMin}(\boldsymbol{\theta}^* , \epsilon)]$ by sampling and counting the frequency of local minima. To be specific, we first choose a parameter point $\boldsymbol{\theta}^*$ randomly to obtain the QNN output state $|\psi^*\rangle=|\psi(\boldsymbol{\theta}^*)\rangle$ and specify the desired overlap $p$ and the error tolerance $\epsilon$. Then, we generate a Haar-random state $|\psi^\perp\rangle$ within the orthogonal complement of $|\psi^*\rangle$ as the unknown component and perform a superposition according to Eq. (4) to obtain a single target state sample $|\phi\rangle$. Then, for each sample, we calculate the gradient and Hessian matrix to check whether they satisfy the condition of local minimum in Eq. (5). After generating $200$ samples, we count how many times the condition of local minimum is satisfied and regard the frequency as an estimation of the probability. The detailed source code used in numerical experiments is available in the supplementary materials for your convenience. We will also describe this clearly in more details in the revision.

---

> > ### Comment · Reviewer_atEi · 2023-08-20
> >
> > Thank you for your response that helps me better understand, I have no further questions.

---

### Official Review · Reviewer_HYxZ · 2023-07-05

**Soundness:** 4 excellent
**Presentation:** 3 good
**Contribution:** 3 good
**Rating:** 7
**Confidence:** 3

**Summary:**

The paper studies parameterized quantum circuits (aka QNNs). These architectures face trainability issues (e.g. barren plateaus) as the number of qubit grows, and several approaches have been explored to mitigate these. The paper analyzes these strategies using the task of training a circuit to transform |0> input state into a desired output state, and provides theoretical and numerical evidence of difficulty of training the circuit.

**Strengths:**

The paper adds new, original result to an important, open problem of training QNNs. The result provide scaling law of the probability of avoiding local minima irrespective of techniques such as special initialization strategy. Importantly, the result explicitly involves the precision parameter, which is one of crucial differentiators between classical networks and QNNs that necessarily use quantum measurement.

Theoretical results are followed with extensive numerical simulations confirming the results in practice.

**Weaknesses:**

The paper is focused on a specific loss, specific task, which leads e.g. to a specific type of dependence of local minima on parameter count (e.g. discussion in lines 206-209 on pg. 6). It is not fully clear how insights from this task translate to other losses/tasks.

The result showing that the number of local minima decreases with the expressibility of the circuit is in line with previous work, but those earlier results are not discussed in detail in the discussion. E.g., Larocca et al. Theory of overparametrization in quantum neural networks. 2021 [Ref 57] is cited in the introduction but not in discussion.

**Questions:**

Discussion in 3.3 mentions complementarity of the abundance of local minima and barren plateaus on the expressibility axis. How do recent results on mitigating barren plateaus via architectural choices instead of initialization (e.g. Wang et al., ICLR'23) affect this understanding?

**Limitations:**

There does not seem to be negative social impact of this theoretical research.

---

> ### Author Rebuttal · Authors · 2023-08-09
>
> We are very grateful for the reviewer's recognition of our work as a novel and technically solid paper with good impact. A detailed response to the reviewer's comments and questions is provided below in a point-by-point manner.
>
> >$\textbf{Comment 1:}$ ``The paper is focused on a specific loss, specific task, which leads e.g. to a specific type of dependence of local minima on parameter count (e.g. discussion in lines 206-209 on pg. 6). It is not fully clear how insights from this task translate to other losses/tasks.''
>
> $\textbf{Re 1:}$ Thanks a lot for the comment. In the main text, we focus on quantum state learning tasks with the fidelity loss function of pure states for the sake of simplicity and ease of understanding. However, the mathematical tools we developed in Appendix A can be directly applied to other scenarios tasks. First, we show similar results for the energy loss function (the expectation value of a Hermitian operator, or say the local loss function) in variational quantum eigensolvers or combinatorial problems in Appendix C. Second, utilizing Lemmas S3 and S7 in Appendix A, similar results can be carried out for the case of mixed states or noisy states by calculating the subspace Haar integration. The loss function can be written as $\mathcal{L}(\boldsymbol{\theta})=1-\langle\phi|\rho(\boldsymbol{\theta})|\phi\rangle$ where $\rho(\boldsymbol{\theta})$ is the noisy output state of QNN and $|\phi\rangle$ is the target state. This generalization will be added in the revised version of our manuscript.
>
> Extending the results to other loss functions, such as alternative distance measures between quantum states, is not as straightforward and requires more careful proof. However, here we can provide a general statement of the insight behind the rigorous results above. The loss value below the average level indicates that the current parameters are relatively good. At this point, moving in a random optimization direction is likely to be less beneficial than staying in place, suggesting the presence of a local minimum. The probability of avoiding such a local minimum should be proportional to the ratio of the number of selected directions (training parameters) over the total number of all possible directions (Hilbert space dimension). Hence, the probability should decay exponentially with the number of qubits, while grows polynomially with the number of training parameters.
>
> >$\textbf{Comment 2:}$ ``The result showing that the number of local minima decreases with the expressibility of the circuit is in line with previous work, but those earlier results are not discussed in detail in the discussion. E.g., Larocca et al. Theory of overparametrization in quantum neural networks. 2021 [Ref 57] is cited in the introduction but not in discussion.''
>
> $\textbf{Re 2:}$ Great thanks for drawing our attention to the literature that we had overlooked in the discussion. Indeed, we agree that the reduction of local minima with the increasing expressibility here bears some resemblance to Ref. [1]. Both of them reflect the notion that high-dimensional spaces can aid optimization. However, due to differences in settings, in Ref. [1], the reduction of local minima occurs after a computational phase transition at $M_c$ upon reaching over-parameterization. In contrast, our work implies that this reduction occurs from the beginning of increasing expressibility in an average sense. We will include this discussion in the revision.
>
> [1] Martin Larocca, Nathan Ju, Diego García-Martín, Patrick J. Coles, and M. Cerezo. Theory of overparametrization in quantum neural networks, 2021.
>
> >$\textbf{Comment 3:}$ ``Discussion in 3.3 mentions complementarity of the abundance of local minima and barren plateaus on the expressibility axis. How do recent results on mitigating barren plateaus via architectural choices instead of initialization (e.g. Wang et al., ICLR'23) affect this understanding?''
>
> $\textbf{Re 3:}$ Thanks for the very insightful question. The complementarity of barren plateaus and local minima discussed here is based on two assumptions, respectively. (a) The QNN is randomly initialized and deep enough to form a unitary $2$-design, so that the Haar integration can give rise to exponentially vanishing mean value and variance of gradients. (b) No prior information is known about the target state except for the loss value $\mathcal{L}^*<\mathcal{L}_c$, so that the subspace integration can give rise to a locally minimal landscape with a probability exponentially close to $1$. A high-overlap initial guess may break assumption (a) but not (b), so it would still encounter local minima, as we discussed in our paper. On the other hand, a good choice of circuit architecture which takes advantage of the specific prior information of the target state, such as the symmetry of the transverse field Ising model considered in Ref. [1], could simultaneously break both assumptions (a) and (b). Thus, problem-inspired architectural choices as in Ref. [1] are likely to not only mitigate barren plateaus, but also reduce the local minima caused by the lack of prior information. We agree that the recent progress on symmetric pruning scheme will shed new lights on the study of QNN. In the revised version, we will elaborate more discussions on recent related papers.
>
> [1] Wang, X., Liu, J., Liu, T., Luo, Y., Du, Y. and Tao, D., 2022, September. Symmetric Pruning in Quantum Neural Networks. In The Eleventh International Conference on Learning Representations (ICLR 2023).

---

> > ### Comment · Reviewer_HYxZ · 2023-08-21
> >
> > Thank you for your response, I have no further questions.

---

### Official Review · Reviewer_oE2w · 2023-07-06

**Soundness:** 3 good
**Presentation:** 3 good
**Contribution:** 2 fair
**Rating:** 6
**Confidence:** 2

**Summary:**

This paper investigates the learnability of the QNN in the task of quantum state learning from a statistical perspective. The paper develops a no-go theorem that proves that when the loss function value is lower than a critical threshold, the probability of avoiding local minima decreases exponentially with the number of qubits, while only increasing polynomially with the circuit depth. Moreover, the paper conducts some numerical experiments to validate the proposed theorem.

**Strengths:**

1. The paper studies a novel research problem, namely the limitation of quantum neural networks in quantum state learning tasks, and analyzes the influence of loss function value information on training difficulty from a statistical perspective, which has certain value for understanding and improving the principles and methods of quantum machine learning.

2. It provides a rigorous and quantitative theoretical analysis showing that when the loss function value is below a critical threshold, the probability of avoiding local minimum decay exponentially with the number of qubits, but only grows polynomially with the circuit depth, revealing the tradeoff between learnability and scalability of QNNs.

**Weaknesses:**

1. The theoretical analysis is only applicable to pure state learning tasks, and in fact, mixed states or noisy states may be encountered in quantum machine learning, so the conclusions and methods may need further generalization and verification.

2. Numerical experiments only use one kind of QNN but do not consider other possible circuit structures and parameterization methods, the results may have certain biases and limitations.

**Questions:**

1. Why choose the ALT structure as the research study in the numerical experiment?  ALT in fact provably does not suffer from the vanishing gradient problem, and does this affect the experimental results as well as the theoretical proof?

2. The abstract mentioned that "the results hold for any circuit structure", and is there any theoretical or experimental proof of this?

**Limitations:**

The paper discusses the limitations and addresses them.

---

> ### Author Rebuttal · Authors · 2023-08-09
>
> We appreciate the reviewer's positive assessment on the correctness, novelty and value of our work. We also thank the reviewer for the helpful feedback. Below is our point-by-point response to the comments and questions.
>
> > $\textbf{Comment 1:}$ ``The theoretical analysis is only applicable to pure state learning tasks, and in fact, mixed states or noisy states may be encountered in quantum machine learning, so the conclusions and methods may need further generalization and verification.''
>
> $\textbf{Re 1:}$ Many thanks for raising this highly practical issue. In the main text, we focused on the case of pure states for the sake of simplicity and ease of better understanding quantum state learning. However, the mathematical tools we developed in Appendix A can be directly applied to the scenario of mixed states or noisy states. For example, suppose that the output state of the QNN is $\rho(\pmb{\theta})$ and the target state is $|\phi\rangle$. The loss function can be defined as the fidelity distance $\mathcal{L}(\pmb{\theta})=1-\langle\phi|\rho(\pmb{\theta})|\phi\rangle$. Utilizing Lemmas S3 and S7, similar results can be carried out by calculating the subspace Haar integration. We shall add this generalization in the revised version of our manuscript.
>
> > $\textbf{Comment 2:}$ ``Numerical experiments only use one kind of QNN but do not consider other possible circuit structures and parameterization methods, the results may have certain biases and limitations.''
>
> $\textbf{Re 2:}$  Many thanks for this helpful comment! Theoretically, our theorem hold for any circuit structure since the proof does not involve specific QNN structures (see our response to the question below for detailed explanation). However, to make the numerical experiments more convincing independently, we agree with your suggestion and will conduct additional experiments using different and common circuit structures and add them in the revised version of our manuscript. Thanks for the suggestion again and we believe considering other possible circuit structures will strengthen our manuscript.
>
> > $\textbf{Comment 3:}$ ``Why choose the ALT structure as the research study in the numerical experiment? ALT in fact provably does not suffer from the vanishing gradient problem, and does this affect the experimental results as well as the theoretical proof?''
>
> $\textbf{Re 3:}$ Thanks a lot for this good question. We choose the ALT structure because it is one of the most extensively used and studied ansatz in variational quantum algorithms [1]. To the best of our knowledge, the ALT ansatz does have better trainability than the hardware-efficient ansatz used in the original paper of barren plateaus. Shallow ALT circuits indeed do not suffer from the vanishing gradient problem. However, if one consider deep circuits to extend the expressibility of QNNs, random QNNs of the repeated-layer type will in general approximate unitary 2-designs given a sufficiently large depth [2], including the ALT ansatz. Thus, the deep ALT ansatz will also suffer from the vanishing gradient problem. Putting aside the points on ALT mentioned above, the specific circuit structure does not affect our theoretical proof (see our response to the next question for detailed explanation), though it may slightly affect the experimental results due to the difference of trainability.
>
> [1]. Nakaji, Kouhei, and Naoki Yamamoto. Expressibility of the alternating layered ansatz for quantum computation. Quantum 5:434, 2021.
>
> [2]. Jonas Haferkamp. Random quantum circuits are approximate unitary t-designs in depth $o(nt^{5+o(1)})$. Quantum, 6:795, 2022.
>
>
> > $\textbf{Comment 4:}$ `` The abstract mentioned that "the results hold for any circuit structure", and is there any theoretical or experimental proof of this?''
>
> $\textbf{Re 4:}$  This is a very insightful question. To provide a clear explanation to this question, we would like to start by reviewing how previous results on QNN trainability depends on the circuit structure. For example, in the original paper of barren plateaus [1], one of the hypotheses is that the random initialized QNN approximates a unitary 2-design so that the Haar integration in the calculation could be carried out safely. This means that their conclusion only applies to random initialized deep QNNs while shallow ones are not subject to such limitations. On the other hand, there is also some literature that does not make any assumption on the circuit structure and initialization of parameters [2], where the ensemble comes from the randomness of unknown learning targets, as in the setup of the Hayden-Preskill thought experiment [2]. The setting in our paper resembles the latter. We suppose that no more information about the target state is known except the measured loss value, so that we can take averages over the orthogonal complement space reasonably. Our theoretical proof does not involve any specific circuit structure and the only relevant hyper-parameters are the qubit count, the circuit depth and the value of quantum Fisher information. Therefore, our results naturally hold for any circuit structure.
>
> [1] Jarrod R. McClean, Sergio Boixo, Vadim N. Smelyanskiy, Ryan Babbush, and Hartmut Neven. Barren plateaus in quantum neural network training landscapes. Nature Communications, 9(1):1–7, mar 2018.
>
> [2] Zoë Holmes, Andrew Arrasmith, Bin Yan, Patrick J. Coles, Andreas Albrecht, and Andrew T. Sornborger. Barren plateaus preclude learning scramblers. Physical Review Letters, 126(19), sep 2020.

---

> > ### Comment · Reviewer_oE2w · 2023-08-20
> >
> > Thank you for your detailed reply that solved my concerns. I have no more questions and adjusted the score (5->6).

---

### Official Review · Reviewer_a4pD · 2023-07-11

**Soundness:** 4 excellent
**Presentation:** 4 excellent
**Contribution:** 3 good
**Rating:** 8
**Confidence:** 4

**Summary:**

This paper introduces a new statistical analysis for training variational quantum circuits (e.g., in terms of quantum neural networks), where the alternating-layered ansatz (Nakaji et al.; ALT), also referred to as the entanglement circuit for quantum ML in Chen et al. 2019 [4], has been characterized as a general quantum state learning task.

In general, the presentation flow is quite good, including a proper introduction to the $l_p$ norm and Dirac notations. The authors have made considerable efforts to guide the readers from the existing learning definition in vector-to-vector mapping to standard encoding based parameterized quantum state learning.

Although some recent works on error analysis in quantum circuit learning [1] and classical encoding circuits [2,3] are unfortunately omitted, the paper actually provides a careful and detailed review of related work in the appendix.

In general, while the theorem is neat, moving from elaborating fidelity loss to fisher information based bound analysis, this paper also conducts a solid local minima analysis. Despite the fact that learnability is considered a no-go perspective and some related work (e.g., [1]) is missing, I believe the theoretical findings and its supporting numerical results conducted good takeaways to the community.

In general, I tend to accept this paper.

***

**References**

1. "Theoretical error performance analysis for variational quantum circuit based functional regression." J Qi et al. npj Quantum Information 9.1 (2023):. Nature
2. "Quantum Circuit Learning," K. Mitarai et al., Physical Review A 98.3 (2018): 032309.
3. "Quantum machine learning in feature hilbert spaces, M. Schuld, Physical Review Letters"
4. "Variational Quantum Circuits for Deep Reinforcement Learning" 2019

**Strengths:**

- The paper provides a clear characterization of the curvature of local minima, which is important to understand the sensitivity of output state with respect to QNN (VQC learning) parameters.

- It provides quantitative limits on good initial guesses related to no free lunch (NFL) theories and adaptive methods for improving the learnability and scalability of QNNs.

- Good presentation quality.

- The paper suggests that no single QNN is universally the best-performing model for learning all target quantum states. This introduces additional complexity for practical applications as it may necessitate more structured QNN architectures and innovative optimization tools.

**Weaknesses:**

- the ensemble setting in Appendix A. 1 is not clear on the motivation of using model ensemble.
- despite the results, there is a level of uncertainty remaining as the exact scaling of QNN depth needed to form a subspace 2-design is not very clear

**Questions:**

1. the ensemble setting in Appendix A. 1 is not clear on the motivation of using model ensemble.

**Limitations:**

- The no-go theorem, while important for understanding the limitations of QNNs, might be a potential barrier to the application of quantum neural networks in real-world scenarios.

- The paper implies that significant future progress will be needed, potentially borrowing insights from the field of deep learning, to overcome the limitations of current QNNs.

---

> ### Author Rebuttal · Authors · 2023-08-09
>
> We greatly appreciate the reviewer's recognition of our work as technically solid, well written, and valuable to the community. We are also grateful to the reviewer for drawing our attention to the literature that we had overlooked. We will ensure to include these references in the revised version of the manuscript. The following is our point-by-point response to your comments and questions.
>
> > $\textbf{Comment 1:} $``The ensemble setting in Appendix A.1 is not clear on the motivation of using model ensemble.''
>
> $\textbf{Re 1:}$ We thank the reviewer for this comment on the ensemble setting. The motivation of our ensemble setting originates from the empirical observation that the probability of encountering local minima when running a variational quantum algorithm depends on the loss value (i.e. the fidelity in quantum state learning tasks). Especially for adaptive methods which are usually not covered by common trainability analyses, the training process often tends to stagnate when a certain loss value is reached. This phenomenon motivates us to study the learnability at a certain loss value. In other words, we want to explore the average learnability among the quantum states $|\phi\rangle$ which have the same fidelity $p$ with the current learned state $|\psi\rangle$, i.e., $|\langle\phi|\psi\rangle|=p$, where our ensemble naturally arises. Graphically, this ensemble forms a circle on the Bloch sphere centered $|\psi\rangle$ as shown in Fig. 1(b) in the manuscript. Mathematically abstracting and generalizing this set of ``quantum states with fixed-fidelity'' leads to the ensemble setting introduced in Appendix A.1.
>
> > $\textbf{Comment 2:} $``Despite the results, there is a level of uncertainty remaining as the exact scaling of QNN depth needed to form a subspace 2-design is not very clear.''
>
> $\textbf{Re 2:}$ Thanks for this insightful comment. As mentioned in the outlook section, although it is well known that a local random quantum circuit of polynomially depth forms an approximate unitary 2-design [1], it remains an open question what the scaling of the depth is to constitute a subspace 2-design. But we can still provide some qualitative analysis. First, the scaling is at most exponential in the number of qubits because the Haar measure over the whole Hilbert space can directly induce a subspace 2-design by constraining the space. Moreover, whether a random QNN can form a subspace 2-design strongly depends on the desired loss value. If the desired loss value is too close to the theoretical minimum, it will be extremely hard for a random QNN to cover those states with such a loss value so that the depth needed will be very large. On the other hand, if the loss value is near its average level, e.g. $1/d$ for the fidelity loss function, the required depth will be relatively smaller since there are many states concentrated into this range.
>
> Here we specially clarify that this specific depth scaling will not affect our conclusion in the manuscript since our ensemble takes average over random target states instead of random QNNs. We mention this issue in the outlook section only because if this depth scaling is known, it would allow us to directly apply our mathematical tools to the setting of random QNNs.
>
> [1] Jonas Haferkamp. Random quantum circuits are approximate unitary t-designs in depth $o(nt^{5+o(1)})$. Quantum, 6:795, 2022.

---

### Author Rebuttal · Authors · 2023-08-10

Dear PC,

We want to express our sincere gratitude to the Program Committee for their hard work in shaping the conference's scientific program. We would also like to thank all the reviewers for recognizing our work as a novel and technically solid paper and recommending acceptance of our submission.

We appreciate the reviewers' time and efforts in reviewing our work and providing constructive feedback. We have addressed all the comments and questions raised by the reviewers in the rebuttal.

Thank you for considering our work!

Yours Sincerely,

Authors of Paper 11169.

---

### Decision · Program_Chairs · 2023-09-21

**Decision:**

Accept (poster)

**Comment:**

This submission investigates the trainability of the QNN in the task of quantum state learning by characterizing the landscape of the loss function.  Specifically, it provides a no-go theorem that proves when the loss function value is lower than a certain threshold, the probability of avoiding local minima decreases exponentially with the number of qubits, while only increasing polynomially with the circuit depth. There is a consensus on the theoretical contribution of the paper.  Besides the presentation issues mentioned in the rebuttal phase, the authors are also advised to provide further discussions about the paper’s relation to VQE, which is closer to the investigated task than general QNN tasks.  In particular, there is work on the convergence of VQE training in the overparameterized setting, as well as other overparameterization work in both VQE and QNN settings, which is worth discussing.